# TopoOT: Topology-Aware Optimal Transport for Test-Time Anomaly Segmentation

## Abstract

Deep topological data analysis (TDA) offers a principled framework for capturing structural invariants such as connectivity and cycles that persist across scales, making it a natural fit for anomaly segmentation (AS). Unlike threshold-based binarisation, which produces brittle masks under distribution shift, TDA allows anomalies to be characterised as disruptions to global structure rather than local fluctuations. We introduce TopoOT, a topology-aware optimal transport (OT) framework that integrates multi-filtration persistence diagrams with test-time adaptation (TTA). Our key innovation is Optimal Transport Chaining, which sequentially aligns persistence diagrams (PDs) across thresholds and filtrations, yielding geodesic stability scores that identify features consistently preserved across scales. These stability-aware pseudo-labels supervise a lightweight head trained online with OT-consistency and contrastive objectives, ensuring robust adaptation under domain shift. Across standard 2D and 3D anomaly detection benchmarks, TopoOT achieves state-of-the-art performance[1], outperforming the most competitive methods by up to +24.1% mean F1 on 2D datasets and +10.2% on 3D anomaly segmentation benchmarks.

## 1 Introduction

Test-time training (TTT) has emerged as a promising paradigm for adapting models under distribution shift, but most approaches remain limited to entropy minimisation or feature consistency, without structured reasoning about data geometry (Sun et al., 2020; Volpi et al., 2022; Zhang et al., 2022). A central limitation of many existing TTT approaches, particularly in dense prediction tasks, is their reliance on heuristic pseudo-labels or confidence thresholds (Liang et al., 2024; Costanzino et al., 2024a; Zhang et al., 2025), which are non-robust (brittle) under distribution shift. Incorporating explicit structural priors provides a principled way to address this gap. The integration of TDA, which extracts persistent features such as connectivity and holes across scales (Zia et al., 2024), and OT, which provides a principled framework for aligning distributions (Cuturi, 2013; Peyré et al., 2019), has received little attention in this context. AS is a particularly compelling domain in which to explore this integration, because it requires pixel-level localisation of irregular patterns whose connectivity and shape are critical, yet conventional threshold-based binarisation often collapses under shift (Cao et al., 2024). By combining TDA's ability to capture structural persistence with OT's alignment capabilities, TTT can move beyond heuristics and yield more stable and adaptive anomaly delineation.

AS demands fine-grained identification of abnormal regions in test images, typically without access to anomalous training examples (Tao et al., 2022). Most existing methods generate continuous anomaly maps that must be binarised (Cao et al., 2024), but thresholds derived from nominal data are brittle across categories and anomaly types (Tong et al., 2024; Wu et al., 2024; Zhou et al., 2024). Supervised approaches (Baitieva et al., 2024; Hu et al., 2024b; Zhu et al., 2024; Ding et al., 2022) can achieve strong performance but require

---

[1]For reproducibility, our implementation is included with this submission

extensive annotation, which is impractical for rare or heterogeneous anomalies (Xie & Mirmehdi, 2007; Qiu et al., 2019). Unsupervised methods (Guo et al., 2025; He et al., 2024) are trained only on nominal data, rely on static thresholds, and fail to preserve structural consistency under domain shift.

Beyond the reliance on brittle thresholds, current approaches to AS and TTA face several underexplored challenges. First, robustness under distribution shift remains insufficient, benchmarks such as MVTec-AD (Bergmann et al., 2019), VisA (Zou et al., 2022), and Real-IAD (Wang et al., 2024a) often understate the variability of anomalies, yet in practice, even minor domain shifts can cause embeddings or thresholds to fail catastrophically. Second, AS research has concentrated on 2D image settings, leaving structural guidance in 3D anomaly detection and segmentation (AD&S) largely unaddressed (Li et al., 2024), despite its importance in industrial inspection. Third, pseudo-labels used in existing TTT frameworks are often derived from entropy or heuristic criteria, providing no guarantees of structural consistency across runs or domains (Zhao et al., 2024). Finally, while efficiency is critical for deployment, there has been little exploration of methods that simultaneously remove threshold dependence and remain lightweight enough for real-time adaptation.

These gaps underscore the need for a framework that (i) eliminates brittle thresholding, (ii) stabilises noisy structural descriptors, (iii) incorporates explicit priors into TTA, and (iv) extends naturally to 3D settings. We propose **TopoOT**, a framework that stabilises pseudo-labels using multi-scale topological cues via persistent homology and aligns them with OT, providing structure-aware supervision for TTT. Although our experiments focus on AS, we view this task as the most natural and demanding testbed for a first exploration of structurally guided TTT, since anomalies disrupt connectivity, boundaries, and higher-order organisation, precisely the features that TDA and OT are designed to capture. Establishing effectiveness in this setting provides a foundation for broader machine learning tasks where structural stability is critical, including domain adaptation under distribution shift (Dan et al., 2024), weak-signal detection in scientific data, and fine-grained visual analysis (Michaeli & Fried, 2024), where subtle structural cues determine class boundaries (Zia et al., 2024). TopoOT embeds structural alignment into the TTT framework. The key contributions are:

- To overcome threshold brittleness, we introduce an **OT-guided, structure-aware representation** that integrates multi-scale topological cues from PDs. This representation produces pseudo-labels that provide adaptive and data-driven supervision for TTT.

- To stabilise noisy topological descriptors, we propose a novel **OT chaining** mechanism that aligns PDs both within a filtration (*cross-PD*) and across sub- and super-level filtrations (*cross-level*), retaining only consistently transported features and discarding spurious ones.

- To integrate structural priors into TTT, we design a lightweight head trained online with two complementary objectives: **OT-consistency**, which preserves *transport-aligned structures, and contrastive separation,* which sharpens anomalous versus nominal boundaries.

- Our approach is *plug-and-play*, integrating seamlessly with different backbones and extending naturally across modalities, generalising from 2D to 3D AD&S (point clouds and multimodal anomaly detection), where connectivity and shape priors are especially critical.

Across diverse datasets, our design consistently delivers robust and generalisable AS. Evaluated on **5** 2D/3D benchmarks and **7** backbones, TopoOT achieves F1 gains up to **+24.1%** on 2D and **+10.2%** on 3D compared to the existing SOTA. It further generalises across models and domains, surpassing TTT baselines by up to **+4.8%**. The lightweight TTT module of TopoOT remains highly efficient, running at **121** FPS while using only **349** MB of GPU memory.

## 2 RELATED WORK

**Anomaly Detection and Segmentation:** AS under distribution shift is challenging as it requires fine-grained detection without supervision, structural priors that capture meaningful data characteristics, and adaptation to

unseen test-time distributions. Unsupervised AD&S avoids labelled anomalies by learning from nominal data (He et al., 2024). Early reconstruction-based methods used autoencoders (Fang et al., 2023; Park et al., 2024; Zuo et al., 2024; Zhou et al., 2025; Wang et al., 2024b), inpainting (Li et al., 2020; Nakanishi et al., 2022; Zavrtanik et al., 2021b; Pirnay & Chai, 2022; Luo et al., 2024), or diffusion models (Yao et al., 2024a; Fučka et al., 2025; Jiang et al., 2024), but often produced blurry reconstructions or overfit to normal patterns. Feature-based approaches compare embeddings to nominal references (Park et al., 2024; Roth et al., 2022; Defard et al., 2021), or use teacher–student frameworks (Deng & Li, 2024; Rudolph et al., 2023; Zhang et al., 2023; Gu et al., 2024) for inductive bias. Generative priors via normalizing flows (Yao et al., 2024b; Gudovskiy et al., 2022; Lei et al., 2023; Kim et al., 2023) or synthetic anomalies (Aota et al., 2023; Li et al., 2024; Hu et al., 2024a; Chen et al., 2024) improved detection, yet typically lack pixel-level precision. Methods such as PatchCore (Roth et al., 2022) and PaDiM (Defard et al., 2021) leverage pre-trained backbones, but remain threshold-dependent and structurally agnostic.

**Optimal Transport in Vision:** OT has been widely applied in computer vision for distribution alignment (Peyré et al., 2019; Cuturi, 2013; Bonneel & Digne, 2023), including domain adaptation (Ge et al., 2021; Fan et al., 2024; Luo & Ren, 2023), object detection, and image restoration (Adrai et al., 2023). In anomaly detection, (Liao et al., 2025) employed robust Sinkhorn distances for industrial inspection. These works show OT's adaptability for handling domain discrepancies, but they typically operate at the distribution level and do not exploit OT for structured feature selection or test-time supervision. While our approach employs a novel *OT chaining* mechanism, entropically regularised OT helps align PDs through cross-PD filtration to capture feature evolution and cross-level filtration to integrate complementary structures, thereby preserving consistently transported features and discarding spurious ones.

**Topological Priors and Test-Time Training:** TDA, particularly persistent homology (PH), has been applied in medical imaging to capture shape and multi-scale structure (Adcock et al., 2014; Berry et al., 2020; Crawford et al., 2020; Garside et al., 2019; Kanari et al., 2018). Yet most uses are offline and not integrated into adaptive learning (Zia et al., 2024). TTT (Liang et al., 2024; Nado et al., 2020; Kim et al., 2022; Colomer et al., 2023; Nguyen et al., 2023; Khurana et al., 2021) adapts models on-the-fly with self-supervised objectives, and TTT4AS (Costanzino et al., 2024a) extended this idea to AS with heuristic pseudo-labels. However, these lack explicit structural reasoning and remain sensitive to noise.

Our approach combines PH-based filtrations with OT alignment to derive stable pseudo-labels, which then guide a lightweight TTT head. This integration moves beyond heuristic thresholds by embedding structural priors directly into TTA, yielding robust and topologically consistent AS.

# 3 OT-GUIDED TEST TIME STRUCTURAL ALIGNMENT FRAMEWORK

**Problem Formulation:** Conventional AS methods produce a dense anomaly score map and obtain binary masks through thresholds calibrated on nominal validation data (Costanzino et al., 2024a) (e.g., percentile rules). Such thresholds are dataset-specific, fail under distribution shift, and often generate masks that under-cover or over-extend the anomalous region. Moreover, they operate pixel-wise and neglect structural information in the anomaly map. To address these limitations, we represent anomaly maps as persistence diagrams (PDs), which capture multi-scale topological features such as connected components and holes. Figure 1 provides an overview of our proposed TopoOT framework. We then introduce an OT–based scoring scheme that evaluates PDs across filtrations and levels, ranking components by their cross-scale consistency. This formulation replaces fixed thresholding with a structural scoring approach designed to produce more consistent anomaly masks under distribution shift.

Building on this, persistence diagrams derived from sub- and super-level filtrations provide the candidate anomaly structures. We apply OT alignment across filtration levels to retain components that persist with low transport cost, while discarding unstable features (that don't persist across PDs). The ranked components are

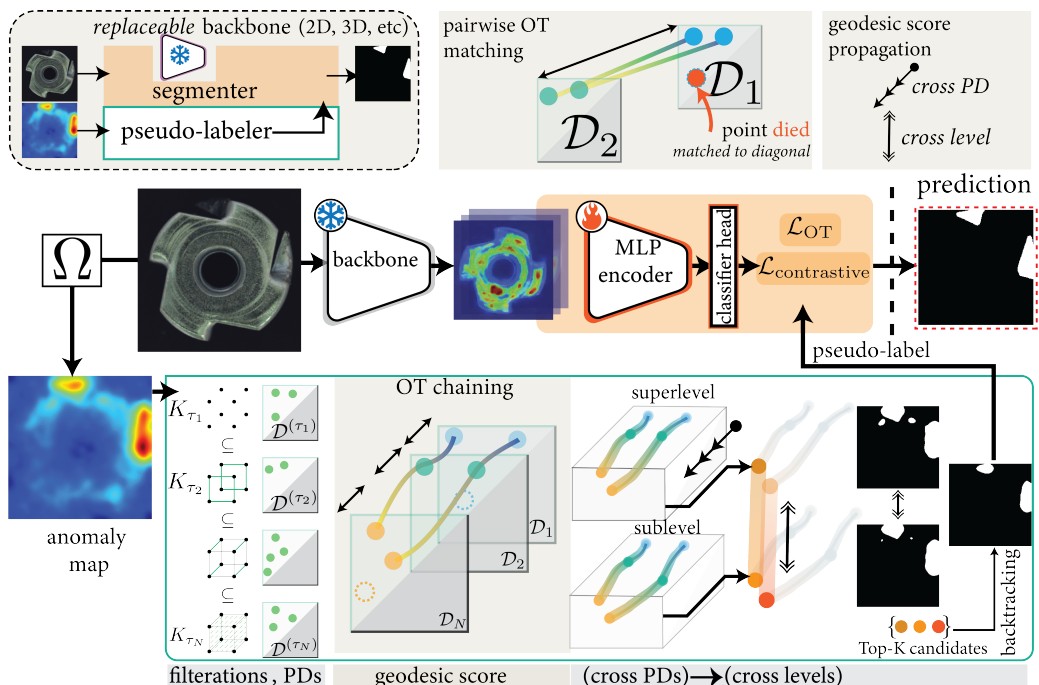

Figure 1: **TopoOT Test-time Training for Anomaly Segmentation.** (Top Left) pipeline simplified view. (Bottom) detailed view. TopoOT replaces conventional thresholding by stabilising anomaly evidence via cross-PD OT matching within each filtration, then fusing sub- and super-level scores with cross-level OT. The resulting global scores yield Top-K pseudo-labels that supervise a lightweight head for final segmentation.

then back-projected into the image domain to form pseudo-labels, which serve as data-dependent supervision at inference in place of fixed thresholds.

During TTT, we keep the replaceable backbone frozen and update only a lightweight head. This head is optimised with two complementary objectives: (i) OT-consistency, which encourages predictions to remain aligned with the stable structures identified by OT, and (ii) contrastive separation, which increases the margin between anomalous and nominal regions. The combination of these objectives yields a segmentation mask that is guided by OT-derived pseudo-labels rather than fixed thresholds.

### 3.1 MULTI-SCALE FILTERING AS FEATURE GENERATION

We start from a continuous anomaly map $A : \Omega \to [0, 1]$ defined over the pixel lattice $\Omega$, same as (Costanzino et al., 2024a). To capture structural variation at multiple thresholds, we fix a sequence of increasing thresholds $\mathcal{T} = \{\tau_1 < \tau_2 < \cdots < \tau_N\}$. For each $\tau_k \in \mathcal{T}$, we define the sublevel and superlevel sets $S_{\tau_k}^{\mathrm{sub}} = \{p \in \Omega : A(p) \leq \tau_k\}$, and $S_{\tau_k}^{\mathrm{sup}} = \{p \in \Omega : A(p) \geq \tau_k\}$. These subsets naturally induce cubical complexes $K_{\tau_k}^{\mathrm{sub}}$, $K_{\tau_k}^{\mathrm{sup}}$, where each cell corresponds to a contiguous block of pixels (a cube in the grid) included whenever its vertices satisfy the relevant threshold condition. The "cubical" construction is appropriate for images/grids, because it respects the pixel adjacency and can be computed efficiently.

By varying the thresholds $\tau_k$, we obtain nested sequences (filtrations) of level sets: $K_{\tau_1}^f \subset K_{\tau_2}^f \subset \cdots \subset K_{\tau_N}^f$, $f \in \{\mathrm{sub}, \mathrm{sup}\}$, where we assume $\tau_1 < \cdots < \tau_N$ for sublevel sets and $\tau_1 > \cdots > \tau_N$ for superlevel

sets. From these filtrations, we compute persistent homology in dimensions $h \in \{0, 1\}$. The result is a persistence diagram $\mathcal{P}_f^h$ at each threshold level. For a filtration $\{K_{\tau_k}^f\}_{k=1}^N$, persistent homology computes, for each dimension $h$, the sequence of homology groups $H_h(K_{\tau_1}^f), \ldots, H_h(K_{\tau_N}^f)$, where $H_h(K_{\tau_k}^f)$ denotes the $h$-th homology group of the complex $K_{\tau_k}^f$. This tracks how $h$-dimensional topological features (connected components for $h = 0$, loops for $h = 1$) appear and disappear along the sequence. A feature $c$ is born at the smallest index $k$ such that it appears in $H_h(K_{\tau_k}^f)$, and dies at the first index $\ell > k$ where it merges into an older feature or becomes trivial. The pair $(b_c, d_c) = (\tau_k, \tau_\ell)$ encodes its lifetime (persistence), and the persistence diagram

$$\mathcal{P}_f^h = \{(b_c, d_c) \mid c \text{ is an } h\text{-dimensional feature in the filtration}\}$$

is the multiset of all such birth–death pairs. For a given threshold $\tau_k$, the diagram $\mathcal{P}_f^h[\tau_k]$, the $H_0$ (homology in dimension 0) captures connected components that are how new components appear (birth) and merge (death) across thresholds. $H_1$ (1-dimensional homology) captures loops or holes (voids), features that appear in superlevel or sublevel sets and disappear at some higher (or lower) threshold. Background on cubical complexes in Appendix A.8.

Each topological feature $c$ in a diagram is represented as a pair $(b_c, d_c)$ of birth and death times; its persistence $\mathrm{pers}(c) = d_c - b_c$ reflects how long it persists. Features with large persistence are more likely to correspond to "meaningful" structural anomalies, while those close to the diagonal (small persistence) are often noise. These ideas align with the discussion review paper by (Zia et al., 2024), which emphasises that PDs and barcodes are robust summaries of topological features of data across scales, invariant to small perturbations, deformation, and noise. The outputs $\{\mathcal{P}_{\mathrm{sub}}^h[\tau_k]\}_{k=1}^N$, and $\{\mathcal{P}_{\mathrm{sup}}^h[\tau_k]\}_{k=1}^N$ serve as multi-scale candidate features. They form the input to the OT-based alignment steps. Rather than acting as direct decision thresholds, these persistence diagrams are treated as a rich feature generation mechanism, capturing anomalies' connected components and holes over multiple scales, which allows the downstream optimal transport stage to judge stability and discriminability among structural candidates.

## 3.2 GEODESIC SCORING OF TOPOLOGICAL FEATURES

The persistence diagrams derived from sub- and super-level filtrations provide a rich but noisy set of candidate features. Many short-lived components arise due to local perturbations in the anomaly map, which, if treated directly, would degrade the reliability of pseudo-labels. A key challenge is how to aggregate these diverse features into a concise set of components that can be meaningfully traced back to the original image. A possible solution is computing a barycenter of diagrams (Turner et al., 2014), but barycenters discard the natural order of filtrations and blur fine-scale structures. Mapping diagrams into kernels or persistence images (Reininghaus et al., 2015) is another alternative, but these yield global embeddings without interpretable correspondences. In contrast, we propose aggregating information by following the flow of diagrams within each filtration sequence using Optimal Transport Chaining. This approach consolidates features into stable representatives for both the sublevel and superlevel filtrations independently, and then fuses the two levels to obtain consensus features.

Formally, let $P = \{p_i = (b_i, d_i)\}_{i=1}^m$ and $Q = \{q_j = (b_j', d_j')\}_{j=1}^n$ be two persistence diagrams, used here as shorthand for $\{\mathcal{P}_f^h[\tau_k]\}$ at different thresholds or filtrations. We define the ground cost as the squared Euclidean distance between pairs of features, and compute the entropic OT plan:

$$\Pi^\star = \underset{\Pi \in \mathcal{U}(P,Q)}{\arg\min} \langle C, \Pi \rangle + \varepsilon H(\Pi) \tag{1}$$

where $\mathcal{U}(P, Q)$ denotes the set of admissible couplings between $P$ and $Q$, and $H(\Pi)$ is the entropy of the transport plan. The regularisation parameter $\varepsilon > 0$ ensures numerical stability and smooth alignments. In

our framework, all transport plans are therefore entropy-regularised Sinkhorn solutions rather than exact Wasserstein couplings because they yield smooth, differentiable, and numerically stable alignments; see Appendix A.6 for further details.

We exploit this transport plan through a novel **OT chaining** mechanism, which consists of two complementary modes: *cross-PD* (intra) filtration and *cross-level* (inter) filtration alignment. In *cross-PD* filtration alignment, OT is applied within a single filtration (sublevel or superlevel) between persistence diagrams at different thresholds $\tau_k$ and $\tau_\ell$. This process identifies features that persist consistently through the filtration, and each candidate $c$ receives a stability score:

$$s(c) = \max_j \frac{\Pi^\star(i(c), j)}{1 + \sqrt{C(i(c), j)}} \cdot \alpha \operatorname{pers}(c) \tag{2}$$

where $\Pi^\star$ is the entropic OT plan between diagrams, $C(i(c), j)$ is the ground cost, $i(c)$ denotes the index of the birth–death pair representing feature $c$ in its persistence diagram, and $\operatorname{pers}(c)$ is the persistence of $c$ as defined in Sec. 3.1. Since $C$ is defined as squared Euclidean distances, we use $\sqrt{C}$ in the denominator to restore a linear distance scale, ensuring that score decay is proportional to distance rather than quadratic. This softens penalisation and allows moderately stable matches to contribute, instead of filtering too aggressively. The maximisation is taken over all possible partners $j$ of candidate $c$ within the filtration, where $j$ indexes features in the comparison persistence diagram. In this way, $s(c)$ reflects the strongest OT-stable match. When points don't get matched between PDs, they are coupled to the diagonal as in standard TDA practice (see Sec. 3.1 and Appendix A.7), ensuring that chain stability scores naturally account for vanishing features. The factor $\alpha \geq 0$ controls the influence of persistence on ranking. Top-$M$ components are selected by maximising stability and persistence and minimising transport cost.

In *cross-level* filtration alignment, we compare candidate sets from the sublevel and superlevel filtrations. Applying OT across sublevel and superlevel filtrations integrates complementary topological cues. Sublevel filtrations emphasise how connected components emerge and merge, while superlevel filtrations highlight how voids and holes evolve. By aligning these perspectives, the method retains structural features that are consistently expressed across both, thereby suppressing spurious components and strengthening anomaly cues. Each candidate $c$ is evaluated with the same stability score $s(c)$ defined above, but here the partner set is drawn from the opposite filtration. This ensures that features are retained only if they exhibit both cross-PD scale persistence and cross-level filtration consistency. The top-$K$ ranked candidates across both filtrations are then collected to form the final set $\mathcal{C}^\star$.

The surviving candidates in $C^\star$ are then projected back to their pixel-level supports on the anomaly map, yielding OT-guided pseudo-labels $\tilde{Y}_{\text{OT}}$. This backprojection is possible because the filtrations in Sec. 3.1 are built by thresholding the anomaly map $A : \Omega \to [0, 1]$ at different anomaly score levels. As these thresholds vary, each retained feature $c \in C^\star$ corresponds to a connected component or hole that remains present in the thresholded maps for all levels between its birth and death $(b_c, d_c)$. Thus $b_c$ and $d_c$ can be interpreted directly as anomaly-score levels at which that structure appears and disappears in the original image. To obtain a pixel-level support for $c$, we choose a representative level near its death time and mark all pixels whose anomaly score exceeds this level. Formally, for each $c \in C^\star$ we define the backprojection threshold as $\tau_{\text{bp}}(c) = d_c$, and the pixel-level support of $c$ as the superlevel set

$$\Gamma(c) = \{\, p \in \Omega : A(p) \geq \tau_{\text{bp}}(c) \,\}. \tag{3}$$

Aggregating the Top-$K$ retained candidates, the OT-guided pseudo-label mask is defined as

$$\tilde{Y}_{\text{OT}}(p) = \mathbf{1}\big(\exists\, c \in C^\star \text{ such that } p \in \Gamma(c)\big), \qquad p \in \Omega, \tag{4}$$

which corresponds to the union of the pixel-level supports of the OT-stable features. For added robustness, one can be a bit conservative when setting the threshold to ensure that the back projected region remains safely within the range where the feature is still present in the filtration. This can be achieved by introducing a small

offset $\delta_{f(c)}$ and thresholding at $\{p : A(p) \geq \max\{0, d_c - \delta_{f(c)}\}\}$, where $f(c) \in \{\text{sublevel, superlevel}\}$ denotes the filtration type. In practice, $\delta_{f(c)}$ is chosen as a small, fixed fraction of the $[0, 1]$ anomaly-score range (e.g., 0.2) and kept constant across all datasets.

These pseudo-labels are inherently multi-scale and data-adaptive, as they emerge from stable OT couplings rather than fixed thresholds. These retained features correspond to connected regions or holes, e.g., defects or gaps, that persist across the filtration process and reflect semantically meaningful structures in the input space. By filtering out noise-induced artefacts, OT alignment produces pseudo-labels that provide robust supervision for TTT.

### 3.3 TopoOT Test-Time Training

The final stage of our pipeline leverages the OT-guided pseudo-labels $\widetilde{Y}_{\text{OT}}$ to adapt the model during inference. Since the backbone feature extractor is frozen, adaptation is performed through a lightweight segmentation head $h_\psi$ attached to the anomaly map representation. This design ensures that the adaptation cost at test time remains negligible, while still allowing the predictions to be tailored to the distribution of the current sample. Training $h_\psi$ is guided by two complementary objectives. First, we introduce an **OT-consistency** loss that encourages the segmentation head $h_\psi$ to reproduce the spatial structures encoded in $\widetilde{Y}_{\text{OT}}$. Given the deviations from the OT-aligned pseudo-labels $\mathcal{L}_{\text{OT}} = \|\widehat{Y} - \widetilde{Y}_{\text{OT}}\|_2$ which enforces consistency with stable transport couplings and prevents overfitting. Second, we incorporate a margin-based contrastive objective to sharpen local decision boundaries in the embedding space produced by $h_\psi$. From the OT-derived pseudo-labels $\widetilde{Y}_{\text{OT}} \in \{0, 1\}^{H \times W}$, we sample pixel pairs $(p, q)$ as similar when $\widetilde{Y}_{\text{OT}}(p) = \widetilde{Y}_{\text{OT}}(q)$ and dissimilar otherwise. Let $z_p, z_q \in \mathbb{R}^D$ denote the L2-normalised embeddings of those pixels. The contrastive loss is:

$$\mathcal{L}_{\text{contrastive}} = (1 - y_{pq}) \|z_p - z_q\|_2^2 + y_{pq} \left[ \max\left(0, \ m - \|z_p - z_q\|_2\right) \right]^2$$

where $y_{pq} \in \{0, 1\}$ encodes dissimilarity and $m > 0$ is a margin. This loss compacts same-label embeddings while enforcing a minimum separation between background and anomalous regions, improving robustness to residual noise in $\widetilde{Y}_{\text{OT}}$. The combined loss is $\mathcal{L}_{\text{TTT}} = \mathcal{L}_{\text{OT}} + \lambda \mathcal{L}_{\text{contrastive}}$ with $\lambda$ controlling the balance between structural consistency and contrastive separation. By optimising $\mathcal{L}_{\text{TTT}}$ on each test sample, the segmentation head $h_\psi$ adapts to dataset-specific distributions without requiring external supervision. The final segmentation mask $\widehat{Y}^{\text{bin}}$ is obtained through a canonical decision rule applied to the adapted predictions of $h_\psi$. Because $h_\psi$ is trained on OT-guided pseudo-labels, this rule is adaptive to each test instance, avoiding dataset-specific calibration and eliminating heuristic threshold tuning.

This *test-time regularisation departs from conventional schemes* in two ways: *(i)* it grounds the adaptation signal in OT-aligned structures, stable across multi-scale filtrations, rather than raw anomaly scores; *(ii)* by integrating contrastive separation, it sharpens class boundaries instead of collapsing toward trivial solutions.

**Stability Observations:** Our evaluation 4 shows that our plug-and-play approach performs consistently well across various backbones and multiple heterogeneous datasets, indicating that cross-PD and cross-level OT chaining yields robust improvements under distribution shift. Appendix A.7 offers an informal OT-based perspective on why cross-PD and cross-level chaining can improve robustness.

## 4 Experimental Setup

**Datasets, Backbones, and Evaluation Protocol**: We evaluate across both 2D and 3D anomaly detection benchmarks. For 2D, RGB datasets **MVTec AD** (Bergmann et al., 2019), **VisA** (Zou et al., 2022), and **Real-IAD** (Wang et al., 2024a) are used with backbones **PatchCore** (Roth et al., 2022), **PaDiM** (Defard et al., 2021), **Dinomaly** (Guo et al., 2025), and **MambaAD** (He et al., 2024). For 3D, we consider multimodal

**MVTec 3D-AD** (RGB + point-cloud) (Bergmann et al., 2021) and pure point-cloud **Anomaly-ShapeNet** (Li et al., 2024), using backbones **CMM** (Costanzino et al., 2024b), **M3DM** (Wang et al., 2023b), and **PO3AD** (Ye et al., 2025). While we report standard anomaly-detection metrics such as image-level AUROC (**I-AUROC**), pixel-level AUROC (**P-AUROC**), and pixel-level AUPRO (**P-AUPRO**) for completeness, our evaluation focuses on pixel-level **Precision**, **Recall**, **F1**, and **IoU** of the final binary masks. AUROC and AUPRO mainly assess ranking quality and can remain high despite poor mask quality under severe pixel imbalance (Bergmann et al., 2019; Zavrtanik et al., 2021a). In contrast, Precision, Recall, and F1 capture the accuracy of detected defect regions, balancing missed detections and false alarms, while IoU offers a stringent measure of spatial overlap (Costanzino et al., 2024a). These metrics align more closely with industrial inspection needs, where the fidelity of the delivered mask is the decisive criterion (Bergmann et al., 2020; Schlüter et al., 2022).

Across both domains, we compare all methods against the TTT baseline **TTT4AS** (Costanzino et al., 2024a). Following **TTT4AS**, we binarise each backbone's AS map at the statistical threshold ($\mu + c\sigma$) and report this variant (**THR**) alongside the **TTT4AS** baseline. All experiments have been conducted on an NVIDIA RTX 5090 GPU with 32GB of VRAM. Detailed hyperparameters and architectural settings are provided in Appendix A.1. TopoOT runs at 121 FPS using 349 MB GPU memory for 2D inference; 3D inference has comparable memory use but lower FPS due to point-cloud operations. Per-dataset timing and memory profiles are given in Appendix A.2.

## 5 RESULTS AND DISCUSSION

We validate TopoOT through analyses: **(i) 2D and 3D AD&S**, benchmarking against state-of-the-art methods; **(ii) Cross Model Domain Adaptation**, where frozen feature extractors are paired with distinct anomaly score maps across 2D and 3D datasets; and **(iii) Ablation Studies**, assessing the contribution of each component. For detailed discussion of limitations and directions for future development, including efficiency tradeoffs and backbone dependency, refer to Appendix A.3.

### 5.1 2D/3D AD&S

We present a comprehensive evaluation of **TopoOT** across five diverse datasets and seven state-of-the-art backbones. The I-AUROCP, P-AUROC, and P-AUPRO metrics are computed directly from each backbone's AS map, while our method operates on the resulting anomaly maps to produce final binary segmentations. The results in Table 1 demonstrate superiority, with **TopoOT** consistently outperforming all baselines. The metrics are the mean per class within each dataset. Our method achieves a **+38.6%** F1 gain over **THR** and **+14.0%** over **TTT4AS** (Costanzino et al., 2024a) on MVTec AD (**PatchCore** (Roth et al., 2022)). For **PaDiM**, it surpasses **THR** by **+20.5%** and **TT4AS** by **+24.1%** . On VisA, it surpasses **TTT4AS** by **+19.7%** (**Dinomaly** (Guo et al., 2025)) and **+8.5%** (**MambaAD** (He et al., 2024)). For Real-IAD, **TopoOT** shows a **+12.3%** and **+11.8%** F1 improvement over **THR**, and a **+21.3%** and **+20.9%** gain over **TTT4AS** for the **Dinomaly** and **MambaAD** backbones, respectively. The advantage extends to 3D, with gains of **+20.7%** (**CMM** (Costanzino et al., 2024b)) and **+24.5%** (**M3DM** (Wang et al., 2023b)) over **THR** on MVTec 3D-AD, alongside **+10.2%** and **+2.2%** improvements over **TTT4AS**. On AnomalyShapeNet (**PO3AD** (Ye et al., 2025)), **TopoOT** also leads with a **+2.9%** and **+1.9%** F1 advantage.

Figure 2 shows that **TopoOT** yields sharper, more semantically coherent segmentations than competing methods. **TopoOT** secures concurrent gains in precision and recall, which in turn increase **IoU**, resulting in consistently superior segmentations across every benchmark. *Per-class quantitative and qualitative results* for each dataset are presented in the Appendix A.4 & A.5. TopoOT consistently achieves sharper boundaries and higher recall across categories. Even in challenging cases like thin or fragmented defects, it remains robust, clearly outperforming other methods across both 2D and 3D domains.

Table 1: Comparison of binary segmentation results. Best results in **bold**; second-best in blue.

| Dataset | Backbone | I-AUROC | P-AUROC | P-AUPRO | TTT Method | Prec. | Rec. | F1 | IoU |
|---------|----------|---------|---------|---------|-----------|-------|------|-----|-----|
| **MVTec AD** (Bergmann et al., 2019) | **PatchCore** (Roth et al., 2022) | 0.991 | 0.981 | 0.934 | **THR** (Roth et al., 2022)
**TTT4AS** (Costanzino et al., 2024a)
**TopoOT** | 0.351
0.388
**0.550** | 0.507
0.648
**0.720** | 0.136
0.382
**0.522** | 0.299
0.293
**0.387** |
| | **PaDiM** (Defard et al., 2021) | 0.979 | 0.975 | 0.921 | **THR** (Roth et al., 2022)
**TTT4AS** (Costanzino et al., 2024a)
**TopoOT** | 0.452
0.330
**0.470** | 0.507
0.579
**0.788** | 0.354
0.318
**0.559** | 0.317
0.274
**0.402** |
| **VisA** (Zou et al., 2022) | **Dinomaly** (Guo et al., 2025) | 0.987 | 0.987 | 0.945 | **THR** (Guo et al., 2025)
**TTT4AS** (Costanzino et al., 2024a)
**TopoOT** | 0.275
0.223
**0.546** | **0.862**
0.811
0.553 | 0.339
0.267
**0.464** | 0.144
0.177
**0.223** |
| | **MambaAD** (He et al., 2024) | 0.943 | 0.985 | 0.910 | **THR** (He et al., 2024)
**TTT4AS** (Costanzino et al., 2024a)
**TopoOT** | 0.200
0.235
**0.416** | 0.785
0.820
0.507 | 0.241
0.289
**0.352** | 0.196
0.145
**0.247** |
| **Real IAD** (Wang et al., 2024a) | **Dinomaly** (Guo et al., 2025) | 0.893 | 0.989 | 0.939 | **THR** (Wang et al., 2024a)
**TTT4AS** (Costanzino et al., 2024a)
**TopoOT** | 0.242
0.154
**0.461** | 0.793
0.801
0.577 | 0.317
0.229
**0.442** | 0.208
0.147
**0.317** |
| | **MambaAD** (He et al., 2024) | 0.863 | 0.985 | 0.905 | **THR** (He et al., 2024)
**TTT4AS** (Costanzino et al., 2024a)
**TopoOT** | 0.188
0.084
**0.305** | 0.653
0.763
0.616 | 0.228
0.137
**0.346** | 0.145
0.080
**0.243** |
| **MVTec 3D-AD** (Bergmann et al., 2021) | **CMM** (Costanzino et al., 2024b) | 0.954 | 0.993 | 0.971 | **THR** (Costanzino et al., 2024b)
**TTT4AS** (Costanzino et al., 2024a)
**TopoOT** | 0.199
0.303
**0.427** | 0.902
0.800
0.845 | 0.275
0.380
**0.482** | 0.232
0.077
**0.343** |
| | **M3DM** (Wang et al., 2023b) | 0.945 | 0.992 | 0.964 | **THR** (Wang et al., 2023b)
**TTT4AS** (Costanzino et al., 2024a)
**TopoOT** | 0.173
0.467
**0.564** | 0.889
0.640
0.767 | 0.245
0.468
**0.490** | 0.232
0.120
**0.364** |
| **AnomalyShapeNet** (Li et al., 2024) | **PO3AD** (Ye et al., 2025) | 0.839 | 0.898 | 0.821 | **THR** (Ye et al., 2025)
**TTT4AS** (Costanzino et al., 2024a)
**TopoOT** | **0.675**
0.562
0.651 | 0.441
0.485
**0.540** | 0.500
0.510
**0.529** | 0.371
0.347
**0.402** |

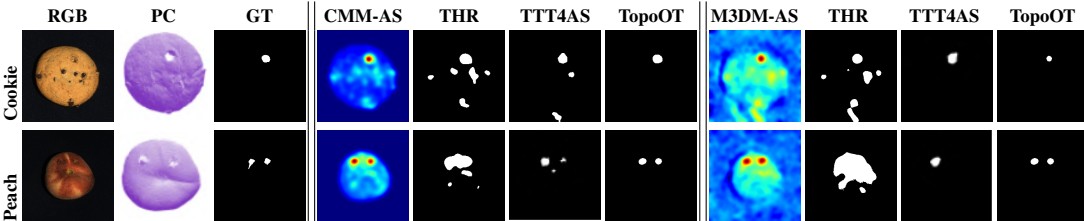

Figure 2: Qualitative comparison of AD&S methods for different objects using the MVTec 3D-AD dataset.

## 5.2 CROSS MODEL DOMAIN ADAPTATION

We validate a plug-and-play transfer strategy that pairs frozen *source* feature extractors with distinct *target* scoring heads across 2D (MVTec, VisA, Real-IAD) and 3D (MVTec-3DAD) domains. As shown in Table 2, the cross-model pipelines preserve topological structure and deliver practical quality without retraining. In 2D, transfers reach F1 up to **0.512** on Real-IAD (PatchCore→MambaAD) and **0.502** on VisA (MambaAD→Dinomaly), with recalls in the **0.71–0.75** band; in 3D, CMM→M3DM offers the highest precision (**0.471**, F1 **0.479**), while M3DM→CMM provides broad coverage (recall **0.791**). Importantly, these domain-adaptation results outperform established baselines across the evaluated datasets, confirming effective cross-model composition and providing a strong substrate for TopoOT to further consolidate gains via stability-aware OT pseudo-labels and adaptive boundary refinement for AS.

## 5.3 ABLATION STUDIES

We validate TopoOT (Table 3). Individual cross-PD filtration alignments yield modest gains. The cross-level filtration alignment is key, providing a larger boost by integrating cross-scale information. The losses $\mathcal{L}_{OT}$ and

Table 2: Cross-model domain adaptation (*features → anomaly scores*).

| Modality | | Dataset | Source → Target | Prec. | Rec. | F1 |
|---|---|---|---|---|---|---|
| 2D | 3D | | (Features → Anomaly Scores) | | | |
| ✓ | | MVTec | PatchCore → PaDiM | 0.419 | 0.673 | 0.430 |
| ✓ | | VisA | MambaAD → Dinomaly | 0.459 | 0.712 | 0.502 |
| ✓ | | Real-IAD | PatchCore → MambaAD | 0.434 | 0.750 | 0.512 |
| | ✓ | MVTec-3DAD | CMM → M3DM | 0.471 | 0.746 | 0.479 |
| | ✓ | MVTec-3DAD | M3DM → CMM | 0.409 | 0.791 | 0.469 |

$\mathcal{L}_{contrastive}$ are effective together, enforcing prediction consistency and feature separation, respectively. Our complete model achieves top performance: **0.522** F1 on PatchCore, **0.482** on CMM, and **0.490** on M3DM.

Table 3: Ablation study showing that combining all OT alignments with losses yields the highest performance.

| TopoOT Components | | | | | 2D-PatchCore | | | 3D-CMM | | | 3D-M3DM | | |
|---|---|---|---|---|---|---|---|---|---|---|---|---|---|
| cross-PD$_{Sub}$ | cross-PD$_{Super}$ | cross-level$_{Sub-super}$ | $\mathcal{L}_{OT}$ | $\mathcal{L}_{contrastive}$ | Prec. | Rec. | F1 | Prec. | Rec. | F1 | Prec. | Rec. | F1 |
| ✓ | | | ✓ | | 0.440 | 0.310 | 0.365 | 0.410 | 0.455 | 0.382 | 0.290 | 0.730 | 0.390 |
| ✓ | | | | ✓ | 0.490 | 0.540 | 0.475 | 0.426 | 0.485 | 0.415 | 0.310 | 0.740 | 0.405 |
| | | ✓ | ✓ | | 0.375 | 0.620 | 0.390 | 0.085 | 0.820 | 0.118 | 0.280 | 0.755 | 0.380 |
| | ✓ | | | ✓ | 0.395 | 0.605 | 0.408 | 0.095 | 0.830 | 0.132 | 0.300 | 0.760 | 0.392 |
| ✓ | ✓ | ✓ | ✓ | | 0.520 | 0.690 | 0.510 | 0.420 | 0.800 | 0.470 | 0.500 | 0.750 | 0.485 |
| ✓ | ✓ | ✓ | | ✓ | 0.510 | 0.680 | 0.505 | 0.405 | 0.770 | 0.460 | 0.490 | 0.740 | 0.475 |
| ✓ | ✓ | ✓ | ✓ | ✓ | **0.550** | **0.720** | **0.522** | **0.427** | **0.845** | **0.482** | **0.564** | **0.767** | **0.490** |

## 6 CONCLUSION

We presented TopoOT, a topology-aware OT framework for anomaly segmentation that replaces brittle thresholding with OT-guided pseudo-labels and stabilises multi-scale persistence features through cross-PD and cross-level filtration chaining. A lightweight head trained with OT-consistency and contrastive objectives enables per-instance TTA that preserves structural stability while sharpening anomaly boundaries. TopoOT achieves SOTA performance on five standard benchmarks, and our theoretical analysis establishes stability and generalisation guarantees.

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

# A    SUPPLEMENTARY MATERIAL

- A.1 outlines the *experimental setup* for 2D and 3D anomaly detection with test-time adaptation and hyperparameter configuration.

- A.2 evaluate the *computational efficiency* of TopoOT by benchmarking its inference time and GPU memory usage in 2D and 3D AS scenarios.

- A.3 discuss *fundamental insights, limitations, and possible extensions* within the context of topological anomaly segmentation.

- A.4 presents *quantitative and qualitative results* on 2D AD&S datasets, including class-wise performance across benchmarks and visual examples that illustrate the effectiveness of OT-guided pseudo-labels.

- A.5 reports *quantitative and qualitative results* on 3D AD&S datasets, covering voxel- and point-cloud modalities, with class-level analysis and qualitative comparisons to baseline methods.

- A.6 recalls *optimal transport preliminaries*, including the 2-Wasserstein distance and its entropy-regularised Sinkhorn variant, and clarifies their role in computing the OT couplings used in our framework.

- A.7 provides a conceptual motivation into optimal transport stability and behaviour.

- A.8 presents the *mathematical formulation* of cubical complex persistence, detailing how primitive cells are hierarchically aggregated to construct filtration levels and ultimately generate persistence vectors that encode topological features.

- A.9 provides a *qualitative analysis* of challenging textural anomaly cases, illustrating how the proposed topology-guided pseudo-labels behave when the backbone anomaly maps exhibit weak topological structure.

- A.10 shows the *ablation study* on the Top-$K$ persistence components, highlighting how varying $K$ impacts the evaluation metrics and that adding lower-ranked components tends to introduce noise and degrade performance.

- A.11 provides the *complete algorithmic pseudocode* for TopoOT, formally defining the multi-scale filtration steps, stability-aware OT chaining, and the spatial backtracking mechanism that drives the test-time adaptation loop.

## A.1    ARCHITECTURAL SETTINGS & HYPERPARAMETERS

**2D Setup.** For all RGB-based AD&S experiments, we employ **DINO** (Caron et al., 2021) as the feature extractor ($F$). Our approach is benchmarked against leading state-of-the-art methods, including the memory-bank based **PatchCore** (Roth et al., 2022), **PaDiM** (Defard et al., 2021), the reconstruction-driven **Dinomaly** (Guo et al., 2025), and **MambaAD** (He et al., 2024). Evaluation is conducted on three widely adopted 2D benchmarks: **MVTec AD** (Bergmann et al., 2019) (15 categories; 3,629 training and 1,725 test images), **VisA** (Zou et al., 2022) (12 objects; 9,621 normal and 1,200 anomalous samples), and **Real-IAD** (Wang et al., 2024a) (30 objects; ~150,000 images in total, comprising 36,465 normal training samples and 114,585 test images with 63,256 normal and 51,329 anomalous). To ensure comparability, all 2D inputs are standardised to a resolution of $224 \times 224$.

**3D Setup.** For multimodal experiments involving RGB and point-cloud modalities, we adopt **DINO-v2** (Oquab et al., 2023) for image features and **Point-MAE** (Zhao et al., 2021) for geometric representations. We benchmark against multimodal memory-bank methods such as **M3DM** (Wang et al., 2023b), as well as reconstruction-oriented baselines including **CMM** (Costanzino et al., 2024b) and **PO3AD** (Ye et al., 2025). The evaluation is performed on two representative 3D benchmarks: **MVTec 3D-AD** (Bergmann et al., 2021)

(10 categories; 2,656 nominal training images and 1,197 test samples) and **Anomaly-ShapeNet** (Li et al., 2024) (40 synthetic classes; 1,600 samples spanning six anomaly types).

**Test-Time Training.** For adaptation, the pretrained backbones are kept frozen while a lightweight MLP head $h_\psi$, consisting of three linear layers with GELU activations, is fine-tuned. The optimisation objective combines an OT-consistency loss ($\epsilon = 0.05$, up to 200 iterations) with a contrastive loss (margin $= 0.4$), balanced equally with weights $\alpha = \lambda = 0.5$. Adaptation proceeds for 5 epochs using the Adam optimiser with a learning rate of $10^{-3}$. Each test sample is processed independently with an effective batch size of one.

## A.2 COMPUTATIONAL COMPLEXITY AND EFFICIENCY

A central strength of the proposed **TopoOT** framework lies in its ability to balance computational complexity with practical efficiency. When evaluated on a single modern GPU, the *complete end-to-end* TopoOT pipeline operates at approximately **2.90 FPS**, while the lightweight **TTT** module alone achieves **121 FPS**. Notably, the **OT** and **TDA** components currently run exclusively on the **CPU**, which constrains the overall end-to-end throughput, while requiring only **349 MB** of GPU memory. This lightweight profile is markedly lower than that of many SOTA anomaly detection baselines. For context, a standard 2D baseline model (Roth et al., 2022) reports an inference time of 0.22 seconds per image, while in the 3D domain, the M3DM (Wang et al., 2023b) model requires 2.86 seconds per image and consumes 6.52 GB of GPU memory. The CMM (Costanzino et al., 2024b) model, though faster at 0.12 seconds per image, still uses 427 MB of memory, **TopoOT** delivers a 14.5× speedup over CMM. In contrast, **TopoOT** not only achieves a significantly higher frame rate but also maintains a highly competitive memory footprint, underscoring its deployability in scenarios where throughput and hardware constraints are decisive.

The breakdown of computational cost, analysed per module, indicates that the construction of cubical complexes and persistence diagrams constitutes the most demanding stage, requiring approximately **0.33 seconds** per sample when aggregated across all complexes. Despite this initial overhead, the subsequent topological alignment stages remain highly efficient: the *intra-level OT* block requires only **5.5 ms** in aggregate, while the *inter-level OT* block converges nearly instantaneously, below **0.05 ms** per alignment. These operations stabilise and align persistence features without imposing a significant runtime burden. Finally, the downstream multilayer perceptron (MLP) classifier adds only **8.3 ms** per evaluation, rendering its contribution negligible.

Table 4 summarises the per-sample runtime for each backbone, split into backbone inference, persistence diagram (PD) computation, OT alignment and the TopoOT TTT head. The PD stage is the main overhead, while OT and TTT are negligible (the TTT head adds only 0.008 s), so the overall end-to-end latency remains comparable to or better than existing 2D/3D anomaly detection baselines.

Table 4: Backbone processing time, TTT method time per sample, and total time (all in seconds).

| | Backbone | | TopoOT | | | | Total | |
|---|---|---|---|---|---|---|---|---|
| **Method** | **Inference Time (s)** | **Memory (GB)** | **PD (s)** | **OT (s)** | **TTT (s)** | **Memory (GB)** | **Time (s)** | **Memory (GB)** |
| **PaDiM** (Defard et al., 2021) | 0.950 | 2.100 | 0.325 | 0.005 | 0.008 | 0.349 | 1.288 | 2.449 |
| **Patchcore** (Roth et al., 2022) | 0.223 | 3.450 | 0.331 | 0.006 | 0.008 | 0.349 | 0.568 | 3.799 |
| **M3DM** (Wang et al., 2023a) | 2.862 | 6.520 | 0.349 | 0.006 | 0.008 | 0.417 | 3.225 | 6.937 |
| **CMM** (Costanzino et al., 2024b) | 0.124 | 0.427 | 0.352 | 0.006 | 0.008 | 0.417 | 0.490 | 0.844 |
| **MambaAD** (He et al., 2024) | 0.027 | 1.480 | 0.374 | 0.006 | 0.008 | 0.370 | 0.415 | 1.850 |
| **Dinomaly** (Guo et al., 2025) | 0.041 | 4.320 | 0.392 | 0.006 | 0.008 | 0.370 | 0.447 | 4.690 |
| **PO3AD** (Ye et al., 2025) | 0.294 | 1.950 | 0.397 | 0.006 | 0.008 | 0.496 | 0.705 | 2.446 |

Taken together, the end-to-end evaluation time per sample remains well within practical limits, supporting real-time operation. The combination of **high FPS**, **minimal GPU consumption**, and the bounded cost of topological computations makes **TopoOT** exceptionally well-suited for industrial adoption. Unlike competing methods that often trade accuracy for efficiency, **TopoOT** achieves both, offering a robust and scalable solution for anomaly detection under stringent practical constraints.

### A.3 DISCUSSION, LIMITATIONS, AND FUTURE DIRECTIONS

The results in the main paper and Appendices A.4 A.5 demonstrate that TopoOT provides a principled strategy for replacing non-robust and heuristic thresholding with stability-aware, OT-guided pseudo-labels. By chaining persistence diagrams across filtrations and integrating sub- and super-level information, the framework yields segmentation masks that are both structurally coherent and robust under distribution shift. Consistent gains across 2D and 3D benchmarks confirm that structural alignment is an effective prior for test-time adaptation.

Despite these advances, several limitations remain. First, the approach still depends on the quality of the anomaly score maps produced by frozen backbones. When upstream representations are noisy or poorly transferable, the extracted persistent features may not provide sufficient structural guidance. Second, while the current formulation generalises naturally to both 2D images and 3D point clouds, it does not yet address spatiotemporal settings such as video or dynamic medical imaging, where temporal coherence and evolving anomaly structure are critical. Third, efficiency trade-offs deserve further study, although TopoOT is lightweight relative to baselines, scaling to real-time, high-resolution deployments in safety-critical domains may require additional optimisations.

Future work can address these challenges along several directions. Differentiable approximations of persistent homology offer a path to end-to-end training with topological losses, enabling tighter integration between backbone features and topological stability. Jointly optimising anomaly map generation and topological filtering through self-supervised objectives could mitigate the reliance on noisy upstream scores. Extending the framework to spatiotemporal domains will require evolving persistence diagrams across frames to capture anomaly lifespans and enforce temporal consistency. Finally, incorporating uncertainty-aware filtration strategies, quantifying stability not only by persistence but also by variability across augmentations or agreement with model uncertainty, could provide more reliable predictions in high-stakes applications such as robotics, autonomous driving, and medical diagnostics.

TopoOT establishes a solid foundation for topology-aware adaptation in anomaly segmentation, highlighting how persistent homology and optimal transport can jointly serve as structural alignment mechanisms for adaptive learning. Its current form addresses critical limitations of threshold-based methods, while future developments promise broader applicability and deeper integration with modern representation learning.

### A.4 ADDITIONAL EXPERIMENTS AND RESULTS ON 2D AD&S DATASETS

Table 5 reports the results of PatchCore on the MVTec AD dataset, evaluated using I-AUROC, P-AUROC, and P-AUPRO. These results are reproduced directly using the official implementation provided by the authors.

Table 5: PatchCore (Roth et al., 2022)on MVTec AD: anomaly scores are I-AUROC, P-AUROC, and P-AUPRO.

| Metric | Bottle | Cable | Capsule | Carpet | Grid | Hazelnut | Leather | MetalNut | Pill | Screw | Tile | T-brush | Transistor | Wood | Zipper | Mean |
|---|---|---|---|---|---|---|---|---|---|---|---|---|---|---|---|---|
| | | | | | **PatchCore — Anomaly Scores** (Roth et al., 2022) | | | | | | | | | | | |
| **I-AUROC** | 1.000 | 0.995 | 0.981 | 0.987 | 0.982 | 1.000 | 1.000 | 1.000 | 0.966 | 0.981 | 0.987 | 1.000 | 1.000 | 0.992 | 0.994 | 0.991 |
| **P-AUROC** | 0.986 | 0.984 | 0.988 | 0.990 | 0.987 | 0.987 | 0.993 | 0.984 | 0.974 | 0.994 | 0.956 | 0.987 | 0.963 | 0.950 | 0.988 | 0.981 |
| **P-AUPRO** | 0.961 | 0.926 | 0.955 | 0.966 | 0.959 | 0.939 | 0.989 | 0.913 | 0.941 | 0.979 | 0.874 | 0.914 | 0.835 | 0.896 | 0.971 | 0.935 |

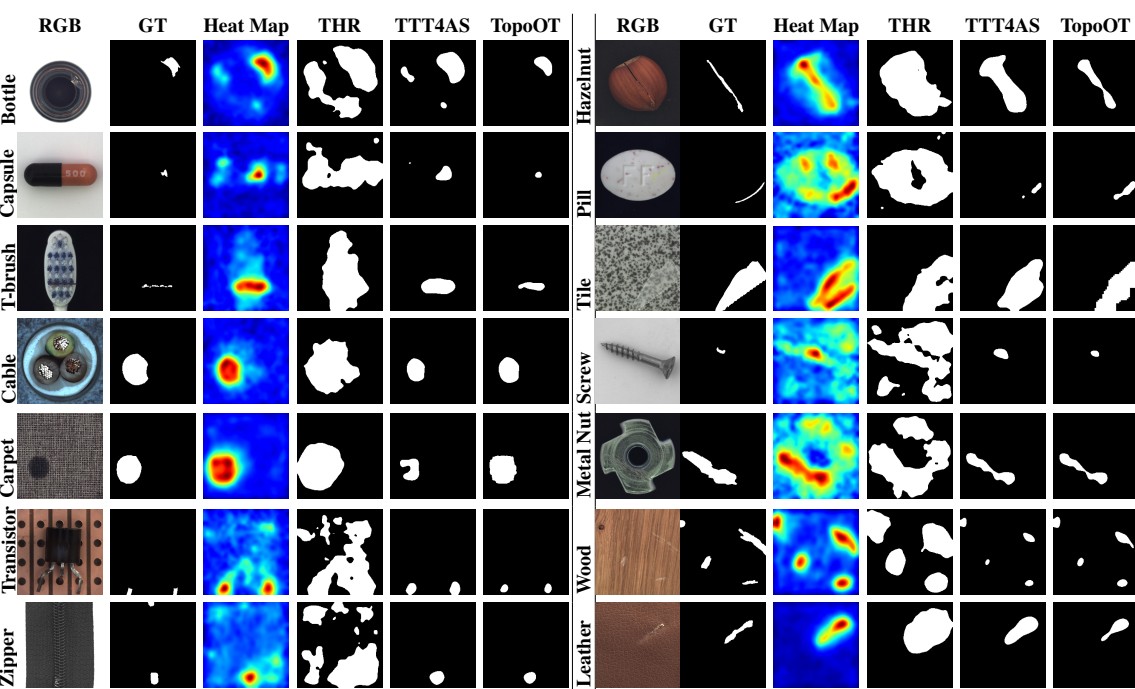

Figure 3: Qualitative comparison of various anomaly detection methods for different objects using PatchCore model on 2D MvTec AD dataset.

Table 6: Performance of PatchCore (Roth et al., 2022) on MVTec AD's 15 categories, comparing binary map strategies: THR ($\mu + 3\sigma$), TTT4AS, and TopoOT. Top results per metric are in **bold** (best) and blue (second-best).

| Metric | Bottle | Cable | Capsule | Carpet | Grid | Hazelnut | Leather | MetalNut | Pill | Screw | Tile | T-brush | Transistor | Wood | Zipper | Mean |
|---|---|---|---|---|---|---|---|---|---|---|---|---|---|---|---|---|
| (a) PatchCore - Binary Map - THR ($\mu + 3\sigma$) (Roth et al., 2022) | | | | | | | | | | | | | | | | |
| Precision | 0.397 | 0.344 | 0.278 | 0.362 | **0.432** | 0.405 | 0.297 | 0.435 | 0.347 | **0.298** | 0.403 | 0.286 | 0.334 | 0.384 | 0.268 | 0.351 |
| Recall | 0.510 | 0.465 | 0.626 | 0.522 | 0.428 | 0.380 | 0.542 | 0.566 | 0.618 | 0.522 | 0.517 | 0.542 | 0.287 | 0.469 | 0.605 | 0.507 |
| F1 Score | 0.175 | 0.194 | 0.085 | 0.092 | 0.078 | 0.120 | 0.045 | 0.311 | 0.188 | 0.066 | 0.209 | 0.123 | 0.114 | 0.121 | 0.119 | 0.136 |
| IoU | 0.310 | 0.334 | 0.222 | **0.407** | 0.283 | 0.367 | 0.262 | 0.316 | 0.287 | 0.202 | 0.179 | 0.262 | 0.238 | 0.297 | **0.513** | 0.299 |
| (b) PatchCore - Binary Map - TTT4AS (Costanzino et al., 2024a) | | | | | | | | | | | | | | | | |
| Precision | 0.662 | 0.502 | 0.163 | 0.413 | 0.185 | 0.425 | 0.212 | 0.644 | 0.337 | 0.046 | 0.644 | 0.272 | 0.391 | 0.470 | 0.449 | 0.388 |
| Recall | **0.664** | 0.565 | 0.632 | **0.824** | **0.787** | 0.861 | 0.893 | 0.528 | 0.740 | 0.361 | 0.495 | 0.594 | 0.462 | **0.664** | **0.644** | 0.648 |
| F1 Score | 0.593 | 0.480 | 0.197 | 0.457 | 0.272 | 0.499 | 0.286 | 0.482 | 0.358 | 0.078 | 0.474 | 0.301 | 0.318 | 0.464 | 0.469 | 0.382 |
| IoU | 0.358 | 0.393 | 0.166 | 0.379 | 0.243 | 0.418 | 0.208 | 0.276 | 0.264 | 0.124 | 0.404 | 0.234 | 0.192 | 0.360 | 0.370 | 0.293 |
| (c) PatchCore - Binary Map -TopoOT | | | | | | | | | | | | | | | | |
| Precision | 0.850 | 0.673 | 0.399 | 0.625 | 0.370 | 0.487 | 0.392 | 0.717 | 0.416 | 0.282 | 0.713 | 0.390 | 0.581 | 0.595 | 0.765 | 0.550 |
| Recall | 0.555 | 0.672 | 0.772 | 0.685 | 0.741 | 0.869 | 0.909 | 0.709 | 0.787 | 0.890 | 0.643 | 0.647 | 0.496 | 0.579 | 0.640 | 0.720 |
| F1 Score | 0.623 | 0.627 | 0.445 | 0.545 | 0.458 | 0.579 | 0.493 | 0.654 | 0.465 | 0.396 | 0.627 | 0.412 | 0.440 | 0.527 | 0.646 | 0.522 |
| IoU | 0.474 | 0.476 | 0.307 | 0.400 | 0.314 | 0.429 | 0.356 | 0.507 | 0.333 | 0.269 | 0.493 | 0.271 | 0.301 | 0.381 | 0.495 | 0.387 |

Table 6 presents the quantitative comparison of PatchCore on MVTec AD's 15 categories using different binary map strategies. Our proposed method, **TopoOT**, consistently outperforms both the threshold-based approach (THR) and the recent TTT4AS method across all evaluation metrics.

Specifically, in terms of mean performance, **TopoOT** achieves an F1 Score of 0.522, significantly higher than THR (0.136) and TTT4AS (0.382). This corresponds to a relative improvement of **+0.386** over THR and **+0.140** over TTT4AS. Similarly, in terms of Precision, TopoOT improves over THR and TTT4AS by **+0.199** and **+0.162**, respectively. A comparable trend is observed for Recall, where TopoOT provides a gain of **+0.213** over THR and **+0.072** over TTT4AS. Beyond overall averages, significant category-level improvements can also be observed in Table 6.

Overall, these results demonstrate that TopoOT not only delivers significant improvements on key categories but also generalises well, outperforming existing methods across the broader range of datasets included in the MVTec AD benchmark. This consistent performance underscores the robustness and effectiveness of our method in 2D anomaly detection tasks.

Table 7 presents the performance of PaDiM on the MVTec AD dataset, evaluated using I-AUROC, P-AUROC, and P-AUPRO. The reported results are reproduced directly from the official implementation released by the authors.

Table 7: PaDiM (Defard et al., 2021) on MVTec AD: anomaly scores are I-AUROC, P-AUROC and P-AUPRO.

| Metric | Bottle | Cable | Capsule | Carpet | Grid | Hazelnut | Leather | MetalNut | Pill | Screw | Tile | T-brush | Transistor | Wood | Zipper | Mean |
|---|---|---|---|---|---|---|---|---|---|---|---|---|---|---|---|---|
| I-AUROC | 0.971 | 0.982 | 0.974 | 0.979 | 0.995 | 0.991 | 0.965 | 0.942 | 0.995 | 0.972 | 0.961 | 0.929 | 0.973 | 0.984 | 0.957 | 0.979 |
| P-AUROC | 0.983 | 0.967 | 0.985 | 0.991 | 0.973 | 0.982 | 0.992 | 0.972 | 0.957 | 0.985 | 0.941 | 0.988 | 0.975 | 0.949 | 0.985 | 0.975 |
| P-AUPRO | 0.948 | 0.888 | 0.935 | 0.962 | 0.946 | 0.926 | 0.978 | 0.856 | 0.927 | 0.944 | 0.860 | 0.931 | 0.845 | 0.911 | 0.959 | 0.921 |

Table 8: Performance evaluation of PaDiM (Defard et al., 2021) across 15 categories of the MVTec AD dataset and their mean, comparing three binary map strategies: (a) THR ($\mu + 3\sigma$), (b) TTT4AS, and (c) TopoTTA. The table highlights the best result for each Precision, Recall, F1 Score, and IoU in **bold** (best) and blue (second-best).

| Metric | Bottle | Cable | Capsule | Carpet | Grid | Hazelnut | Leather | MetalNut | Pill | Screw | Tile | T-brush | Transistor | Wood | Zipper | Mean |
|---|---|---|---|---|---|---|---|---|---|---|---|---|---|---|---|---|
| **(a) PaDiM - Binary Map - THR ($\mu + 3\sigma$) (Defard et al., 2021)** | | | | | | | | | | | | | | | | |
| Precision | 0.729 | 0.580 | 0.287 | **0.561** | 0.327 | **0.586** | 0.306 | 0.540 | **0.410** | 0.196 | 0.131 | **0.416** | 0.462 | **0.576** | 0.676 | 0.452 |
| Recall | 0.321 | 0.249 | 0.813 | 0.736 | 0.708 | 0.477 | 0.927 | 0.281 | 0.493 | 0.712 | 0.005 | 0.514 | 0.349 | 0.399 | 0.615 | 0.507 |
| F1 Score | 0.343 | 0.280 | 0.325 | 0.523 | 0.407 | 0.433 | **0.396** | 0.292 | 0.337 | 0.295 | 0.009 | 0.391 | 0.307 | 0.375 | 0.596 | 0.354 |
| IoU | 0.310 | 0.290 | 0.330 | 0.340 | 0.320 | 0.300 | 0.310 | 0.330 | 0.320 | 0.300 | 0.280 | 0.350 | 0.330 | 0.310 | 0.335 | 0.317 |
| **(b) PaDiM - Binary Map - TTT4AS (Costanzino et al., 2024a)** | | | | | | | | | | | | | | | | |
| Precision | 0.585 | 0.412 | 0.176 | 0.429 | 0.199 | 0.349 | 0.208 | 0.519 | 0.269 | 0.088 | 0.137 | 0.258 | 0.472 | 0.355 | 0.499 | 0.330 |
| Recall | 0.438 | 0.500 | 0.707 | 0.769 | 0.726 | 0.637 | 0.916 | 0.491 | 0.568 | 0.735 | 0.123 | 0.595 | 0.425 | 0.416 | 0.648 | 0.579 |
| F1 Score | 0.429 | 0.395 | 0.214 | 0.459 | 0.290 | 0.376 | 0.293 | 0.386 | 0.262 | 0.153 | 0.103 | 0.283 | 0.291 | 0.319 | 0.512 | 0.318 |
| IoU | 0.280 | 0.270 | 0.280 | 0.270 | 0.260 | 0.310 | 0.270 | 0.280 | 0.270 | 0.270 | 0.270 | 0.280 | 0.270 | 0.270 | 0.270 | 0.274 |
| **(c) PaDiM - Binary Map - TopoOT** | | | | | | | | | | | | | | | | |
| Precision | **0.750** | **0.648** | **0.355** | 0.523 | 0.463 | 0.358 | 0.246 | **0.574** | 0.307 | 0.266 | 0.685 | 0.268 | **0.492** | 0.439 | 0.678 | 0.470 |
| Recall | **0.689** | **0.670** | **0.828** | 0.942 | 0.805 | 0.885 | 0.987 | 0.636 | 0.783 | 0.905 | 0.742 | 0.920 | 0.547 | 0.756 | 0.724 | 0.788 |
| F1 Score | 0.718 | 0.658 | 0.496 | 0.672 | 0.587 | 0.509 | 0.393 | 0.603 | 0.441 | 0.411 | 0.712 | 0.415 | 0.518 | 0.555 | 0.700 | 0.559 |
| IoU | 0.390 | 0.410 | 0.400 | 0.420 | 0.380 | 0.400 | 0.410 | 0.390 | 0.400 | 0.410 | 0.420 | 0.390 | 0.410 | 0.400 | 0.400 | 0.402 |

Table 8 shows the performance comparison of PaDiM on MVTec AD's 15 categories using different binary map strategies. Our proposed method, **TopoOT**, achieves consistent improvements across all metrics when compared to both THR and TTT4AS.

On average, TopoOT improves the F1 Score by **+0.205** over THR and by **+0.241** over TTT4AS. Similarly, Precision increases by **+0.018** compared to THR and by **+0.14** over TTT4AS. In Recall, TopoOT shows even stronger gains of **+0.281** against THR and **+0.209** over TTT4AS.

Overall, these results confirm that TopoOT not only delivers significant improvements in individual categories but also generalises well across the full MVTec AD benchmark. Its consistent superiority over both threshold-based and test-time training baselines demonstrates the effectiveness of our approach for 2D anomaly detection tasks.

Table 9 presents the results of MambaAD on VisA (12 classes), where I-AUROC, P-AUROC, and P-AUPRO are reported as **mean per class**. The results are reproduced directly using the official implementation provided by the authors.

Table 9: MambaAD (He et al., 2024) on VisA (12 classes), I-AUROC, P-AUROC, P-AUPRO, metrics are **mean per class**.

| Metric | candle | capsules | cashew | chewinggum | fryum | macaroni1 | macaroni2 | pcb1 | pcb2 | pcb3 | pcb4 | pipe_fryum | Mean |
|---|---|---|---|---|---|---|---|---|---|---|---|---|---|
| **I-AUROC** | 0.968 | 0.918 | 0.945 | 0.977 | 0.952 | 0.916 | 0.816 | 0.954 | 0.942 | 0.937 | 0.999 | 0.987 | **0.943** |
| **P-AUROC** | 0.990 | 0.991 | 0.943 | 0.981 | 0.969 | 0.995 | 0.995 | 0.998 | 0.989 | 0.991 | 0.986 | 0.991 | **0.985** |
| **P-AUPRO** | 0.955 | 0.918 | 0.878 | 0.797 | 0.916 | 0.952 | 0.962 | 0.928 | 0.896 | 0.891 | 0.876 | 0.951 | **0.910** |

Table 10: Performance evaluation of MambaAD (He et al., 2024) 12 categories (VisA classes) and their mean, comparing three binary map strategies: (a) THR ($\mu + 3\sigma$), (b) TTT4AS, and (c) OT-TopoTTA. The table highlights the best result for each Precision, Recall, and F1 Score metric in **bold black** and the second-best in blue.

| Metric | candle | capsules | cashew | chewinggum | fryum | macaroni1 | macaroni2 | pcb1 | pcb2 | pcb3 | pcb4 | pipe_fryum | Mean |
|---|---|---|---|---|---|---|---|---|---|---|---|---|---|
| (a) MambaAD - Binary Map - THR ($\mu + 3\sigma$) (He et al., 2024) | | | | | | | | | | | | | |
| Precision | 0.111 | 0.291 | 0.163 | 0.368 | 0.265 | 0.049 | 0.060 | 0.224 | 0.166 | 0.209 | 0.333 | 0.166 | 0.200 |
| Recall | **0.874** | 0.741 | 0.699 | 0.796 | 0.659 | 0.775 | 0.804 | **0.954** | **0.816** | 0.779 | **0.648** | 0.877 | 0.785 |
| F1 Score | 0.172 | 0.357 | 0.174 | 0.468 | 0.207 | 0.088 | 0.104 | 0.278 | 0.255 | 0.299 | 0.396 | 0.092 | 0.241 |
| IoU | 0.105 | **0.259** | 0.105 | 0.334 | 0.127 | 0.048 | 0.058 | 0.278 | 0.255 | 0.299 | **0.396** | 0.092 | 0.196 |
| (b) MambaAD - Binary Map - TTT4AS (Guo et al., 2025) | | | | | | | | | | | | | |
| Precision | 0.185 | 0.389 | 0.229 | 0.335 | 0.263 | 0.079 | 0.052 | 0.239 | 0.235 | 0.243 | 0.398 | 0.178 | 0.235 |
| Recall | 0.807 | **0.879** | **0.857** | 0.867 | 0.716 | 0.807 | 0.858 | 0.889 | 0.824 | **0.822** | 0.601 | 0.918 | 0.820 |
| F1 Score | 0.264 | **0.440** | 0.275 | 0.484 | 0.190 | 0.137 | 0.097 | 0.307 | 0.339 | 0.331 | 0.419 | 0.186 | 0.289 |
| IoU | 0.104 | 0.258 | 0.114 | 0.331 | 0.113 | 0.032 | 0.025 | 0.163 | 0.122 | 0.167 | 0.217 | 0.095 | 0.145 |
| (c) MambaAD - Binary Map - OT-TopoTTA | | | | | | | | | | | | | |
| Precision | 0.311 | 0.483 | 0.336 | 0.577 | 0.444 | 0.198 | 0.184 | 0.507 | 0.431 | 0.480 | 0.702 | 0.341 | 0.416 |
| Recall | 0.542 | 0.460 | 0.563 | 0.573 | 0.290 | 0.565 | 0.664 | 0.529 | 0.410 | 0.469 | 0.317 | 0.696 | 0.507 |
| F1 Score | **0.295** | 0.357 | **0.314** | **0.528** | 0.199 | **0.247** | **0.258** | **0.462** | **0.388** | **0.433** | 0.392 | **0.346** | **0.352** |
| IoU | **0.196** | 0.246 | **0.217** | **0.394** | **0.200** | **0.157** | **0.163** | **0.328** | **0.267** | 0.298 | 0.267 | **0.226** | **0.247** |

Table 10 presents the performance comparison of MambaAD on the VisA dataset across 12 categories, using different binary map strategies. Our proposed method, **TopoOT**, consistently achieves higher scores across Precision, Recall, and F1 compared to THR and TTT4AS.

On average, TopoOT improves the F1 Score by **+0.111** over THR and by **+0.085** over TTT4AS. Similarly, Precision increases by **+0.216** compared to THR and by **+0.193** over TTT4AS. In Recall, TopoOT performed a little low as compared with compared to THR and TTT4AS.

Overall, these results confirm that TopoOT not only achieves notable improvements on specific challenging categories but also generalises effectively across all VisA classes. Its consistent superiority over both traditional thresholding and recent test-time training methods highlights the robustness of our approach for 2D anomaly detection under complex real-world scenarios.

Table 11 reports the results of MambaAD on Real-IAD (30 classes), where anomaly scores are given in terms of I-AUROC, P-AUROC, and P-AUPRO. The results are obtained directly using the official implementation provided by the authors.

Table 12 shows that TopoOT frequently outperforms both THR and TTT4AS, securing the top rank in most metrics across different 30 classes of the Real-IAD Dataset. On average, our TopoOT has an F1 Score of

Table 11: MambaAD (He et al., 2024) on Real-IAD (30 classes). Anomaly scores I-AUROC, P-AUROC and P-AUPRO.

| Metric | audiojack | b-cap | b-battery | e-cap | eraser | f-hood | mint | mounts | pcb | p-battery | p-nut | p-plug | p-doll | regulator | r-base | s-set |
|---|---|---|---|---|---|---|---|---|---|---|---|---|---|---|---|---|
| | | | | | **MambaAD (He et al., 2024) — Anomaly Scores** | | | | | | | | | | | |
| I-AUROC | 0.842 | 0.928 | 0.798 | 0.780 | 0.875 | 0.793 | 0.701 | 0.868 | 0.891 | 0.902 | 0.871 | 0.857 | 0.880 | 0.697 | 0.980 | 0.944 |
| P-AUROC | 0.977 | 0.997 | 0.981 | 0.970 | 0.992 | 0.987 | 0.965 | 0.992 | 0.992 | 0.994 | 0.994 | 0.990 | 0.992 | 0.976 | 0.997 | 0.988 |
| P-AUPRO | 0.839 | 0.972 | 0.862 | 0.894 | 0.937 | 0.863 | 0.726 | 0.935 | 0.931 | 0.953 | 0.961 | 0.915 | 0.954 | 0.870 | 0.988 | 0.894 |

| Metric | switch | tape | t-block | t-brush | toy | t-brick | transistor1 | u-block | usb | u-adaptor | vcpill | w-beads | woodstick | zipper | Mean |
|---|---|---|---|---|---|---|---|---|---|---|---|---|---|---|---|
| | | | | | **MambaAD (He et al., 2024) — Anomaly Scores** | | | | | | | | | | |
| I-AUROC | 0.917 | 0.968 | 0.961 | 0.851 | 0.830 | 0.705 | 0.944 | 0.897 | 0.920 | 0.794 | 0.883 | 0.825 | 0.804 | 0.992 | 0.863 |
| P-AUROC | 0.982 | 0.998 | 0.998 | 0.975 | 0.960 | 0.966 | 0.994 | 0.995 | 0.992 | 0.973 | 0.987 | 0.980 | 0.977 | 0.993 | 0.985 |
| P-AUPRO | 0.929 | 0.980 | 0.982 | 0.914 | 0.863 | 0.747 | 0.965 | 0.954 | 0.952 | 0.825 | 0.893 | 0.845 | 0.827 | 0.976 | 0.905 |

Table 12: Performance evaluation of MambaAD (He et al., 2024) across 30 classes (Real-IAD Dataset) and their mean, comparing three binary map strategies: (a) THR ($\mu + 3\sigma$), (b) TTT4AS, and (c) TopoOT. The best result for each Precision, Recall, and F1 Score is in **bold** and the second-best in blue.

| Metric | audiojack | b-cap | b-battery | e-cap | eraser | f-hood | mint | mounts | pcb | p-battery | p-nut | p-plug | p-doll | regulator | r-base | s-set |
|---|---|---|---|---|---|---|---|---|---|---|---|---|---|---|---|---|
| | | | | **(a) MambaAD - Binary Map - THR ($\mu + 3\sigma$) (He et al., 2024)** | | | | | | | | | | | | |
| Precision | 0.164 | 0.055 | **0.199** | **0.202** | 0.121 | 0.126 | 0.082 | 0.209 | **0.438** | 0.178 | 0.132 | 0.101 | 0.122 | 0.074 | 0.144 | 0.156 |
| Recall | 0.510 | 0.944 | 0.333 | 0.475 | 0.648 | 0.514 | 0.385 | 0.759 | 0.472 | 0.815 | 0.783 | 0.846 | 0.794 | 0.548 | 0.950 | 0.743 |
| F1 Score | 0.210 | 0.100 | 0.160 | 0.181 | 0.188 | 0.178 | 0.120 | 0.254 | 0.309 | 0.280 | 0.202 | 0.173 | 0.189 | 0.107 | 0.227 | 0.245 |
| IoU | 0.133 | 0.055 | 0.102 | 0.116 | 0.114 | 0.112 | 0.076 | 0.162 | 0.212 | 0.171 | 0.124 | 0.100 | 0.114 | 0.062 | 0.139 | 0.155 |
| | | | | **(b) MambaAD - Binary Map - TTT4AS (Costanzino et al., 2024a)** | | | | | | | | | | | | |
| Precision | 0.062 | 0.027 | 0.075 | 0.039 | 0.059 | 0.051 | 0.046 | 0.097 | 0.075 | 0.084 | 0.055 | 0.048 | 0.073 | 0.034 | 0.071 | 0.091 |
| Recall | **0.605** | **0.953** | **0.629** | **0.684** | **0.739** | **0.692** | **0.461** | **0.833** | **0.870** | **0.887** | **0.799** | **0.870** | **0.830** | 0.534 | **0.951** | **0.762** |
| F1 Score | 0.109 | 0.052 | 0.108 | 0.072 | 0.105 | 0.090 | 0.075 | 0.164 | 0.135 | 0.151 | 0.099 | 0.089 | 0.124 | 0.061 | 0.127 | 0.154 |
| IoU | 0.062 | 0.027 | 0.066 | 0.039 | 0.059 | 0.050 | 0.044 | 0.097 | 0.074 | 0.084 | 0.055 | 0.048 | 0.071 | 0.034 | 0.071 | 0.091 |
| | | | | **(c) MambaAD - Binary Map - TopoOT** | | | | | | | | | | | | |
| Precision | **0.239** | **0.245** | 0.164 | 0.156 | **0.254** | **0.183** | **0.133** | **0.422** | 0.297 | **0.404** | **0.269** | **0.229** | **0.324** | **0.171** | **0.411** | **0.398** |
| Recall | 0.491 | 0.829 | 0.513 | 0.612 | 0.585 | 0.544 | 0.355 | 0.637 | 0.745 | 0.679 | 0.695 | 0.766 | 0.584 | **0.451** | 0.803 | 0.629 |
| F1 Score | **0.284** | **0.347** | **0.169** | **0.225** | **0.303** | **0.233** | **0.155** | **0.444** | **0.367** | **0.435** | **0.341** | **0.317** | **0.365** | **0.203** | **0.465** | **0.430** |
| IoU | **0.197** | **0.231** | 0.106 | **0.143** | **0.203** | **0.159** | **0.102** | **0.324** | **0.255** | **0.317** | **0.236** | **0.206** | **0.246** | **0.139** | **0.342** | **0.322** |

| Metric | switch | tape | t-block | t-brush | toy | t-brick | transistor1 | u-block | usb | u-adaptor | vcpill | w-beads | woodstick | zipper | Mean |
|---|---|---|---|---|---|---|---|---|---|---|---|---|---|---|---|
| | | | **(a) MambaAD - Binary Map - THR ($\mu + 3\sigma$) (He et al., 2024)** | | | | | | | | | | | | |
| Precision | 0.252 | 0.129 | 0.165 | 0.396 | 0.149 | **0.264** | 0.218 | 0.131 | 0.289 | 0.053 | 0.331 | **0.194** | 0.188 | 0.378 | 0.188 |
| Recall | **0.736** | **0.953** | **0.951** | 0.442 | 0.535 | 0.268 | 0.729 | 0.794 | 0.699 | 0.586 | 0.598 | 0.429 | 0.540 | 0.805 | 0.653 |
| F1 Score | 0.331 | 0.214 | 0.265 | 0.309 | 0.188 | **0.213** | 0.316 | 0.211 | 0.320 | 0.090 | 0.380 | 0.227 | 0.242 | 0.399 | 0.228 |
| IoU | 0.226 | 0.128 | 0.161 | 0.203 | 0.118 | **0.148** | 0.200 | 0.130 | 0.211 | 0.050 | 0.270 | 0.148 | 0.155 | 0.269 | 0.145 |
| | | | **(b) MambaAD - Binary Map - TTT4AS (Costanzino et al., 2024a)** | | | | | | | | | | | | |
| Precision | 0.123 | 0.085 | 0.063 | 0.246 | 0.053 | 0.078 | 0.120 | 0.071 | 0.086 | 0.024 | 0.169 | 0.099 | 0.085 | 0.221 | 0.084 |
| Recall | 0.856 | **0.956** | **0.989** | 0.624 | 0.636 | 0.585 | 0.893 | 0.838 | 0.907 | **0.577** | 0.741 | 0.626 | 0.668 | 0.905 | 0.763 |
| F1 Score | 0.200 | 0.149 | 0.116 | 0.299 | 0.093 | 0.131 | 0.205 | 0.125 | 0.153 | 0.045 | 0.258 | 0.159 | 0.146 | 0.310 | 0.137 |
| IoU | 0.122 | 0.084 | 0.063 | 0.194 | 0.052 | 0.077 | 0.119 | 0.071 | 0.086 | 0.023 | 0.163 | 0.096 | 0.084 | 0.204 | 0.080 |
| | | | **(c) MambaAD - Binary Map - TopoOT** | | | | | | | | | | | | |
| Precision | **0.430** | **0.404** | **0.423** | **0.454** | **0.222** | 0.189 | **0.348** | **0.338** | **0.370** | **0.127** | **0.481** | **0.263** | **0.258** | **0.535** | **0.305** |
| Recall | 0.661 | 0.747 | 0.866 | 0.409 | 0.594 | 0.510 | 0.573 | 0.721 | 0.736 | 0.572 | 0.510 | 0.476 | 0.521 | 0.665 | 0.616 |
| F1 Score | **0.455** | **0.460** | **0.520** | **0.368** | **0.292** | 0.235 | **0.385** | **0.412** | **0.440** | **0.179** | **0.430** | **0.296** | **0.318** | **0.520** | **0.346** |
| IoU | **0.330** | **0.322** | **0.378** | **0.247** | **0.205** | 0.159 | **0.267** | **0.295** | **0.317** | **0.117** | **0.311** | **0.208** | **0.221** | **0.378** | **0.243** |

**0.346**, showing an improvement of **+0.058** over THR($\mu + 3\sigma$) and an astounding **+0.209** improvement over TTT4AS. Similarly, TopoOT has **0.305** Precision, which is **+0.117** more than the THR and **+0.221** more than TTT4AS. This trend of significant improvement is not limited to a few instances, but our method's consistent performance across the 30 classes shown in the Table 12 indicates its robustness and effectiveness. While the specific percentages of improvement vary, the overall pattern is clear that our method also outperformed on other classes, making TopoOT a highly effective and robust technique for anomaly detection.

Table 13 presents the results of Dinomaly on VisA (12 classes), with anomaly scores reported in terms of I-AUROC, P-AUROC, and P-AUPRO. The results are reproduced directly using the official implementation provided by the authors.

Table 13: Dinomaly (Guo et al., 2025) on VisA (12 classes). Anomaly scores I-AUROC, P-AUROC and P-AUPRO.

| Metric | candle | capsules | cashew | chewinggum | fryum | macaroni1 | macaroni2 | pcb1 | pcb2 | pcb3 | pcb4 | pipe_fryum | Mean |
|---|---|---|---|---|---|---|---|---|---|---|---|---|---|
| **(a) Dinomaly (Guo et al., 2025) - Anomaly Score** | | | | | | | | | | | | | |
| **I-AUROC** | 0.987 | 0.986 | 0.987 | 0.998 | 0.988 | 0.980 | 0.959 | 0.991 | 0.993 | 0.989 | 0.998 | 0.992 | 0.987 |
| **P-AUROC** | 0.994 | 0.996 | 0.971 | 0.991 | 0.966 | 0.996 | 0.997 | 0.995 | 0.980 | 0.984 | 0.987 | 0.992 | 0.987 |
| **P-AUPRO** | 0.954 | 0.974 | 0.940 | 0.881 | 0.935 | 0.964 | 0.987 | 0.951 | 0.913 | 0.946 | 0.944 | 0.952 | 0.945 |

Table 14: Performance evaluation of Dinomaly (Guo et al., 2025) across 12 categories (VisA classes) and their mean, comparing three binary map strategies: (a) THR ($\mu + 3\sigma$), (b) TTT4AS, and (c) TopoOT. The table highlights the best result for each Precision, Recall, and F1 Score metric in **bold black** and the second-best in blue.

| Metric | candle | capsules | cashew | chewinggum | fryum | macaroni1 | macaroni2 | pcb1 | pcb2 | pcb3 | pcb4 | pipe_fryum | Mean |
|---|---|---|---|---|---|---|---|---|---|---|---|---|---|
| **(a) Dinomaly - Binary Map - THR ($\mu + 3\sigma$) (Guo et al., 2025)** | | | | | | | | | | | | | |
| **Precision** | 0.190 | 0.316 | 0.239 | 0.384 | 0.307 | 0.109 | 0.111 | 0.300 | 0.275 | 0.318 | 0.518 | 0.231 | 0.275 |
| **Recall** | **0.908** | **0.936** | 0.824 | **0.889** | 0.740 | **0.947** | **0.970** | 0.862 | **0.847** | **0.861** | 0.674 | 0.885 | **0.862** |
| **F1 Score** | 0.286 | 0.396 | 0.285 | 0.510 | 0.247 | 0.189 | 0.195 | 0.373 | 0.380 | 0.435 | **0.522** | 0.246 | 0.339 |
| **IoU** | 0.116 | 0.230 | 0.108 | 0.289 | 0.093 | 0.034 | 0.032 | 0.146 | 0.126 | 0.176 | 0.309 | 0.069 | 0.144 |
| **(b) Dinomaly - Binary Map - TTT4AS (Costanzino et al., 2024a)** | | | | | | | | | | | | | |
| **Precision** | 0.175 | 0.369 | 0.217 | 0.318 | 0.250 | 0.075 | 0.049 | 0.227 | 0.223 | 0.231 | 0.378 | 0.169 | 0.223 |
| **Recall** | 0.798 | 0.869 | **0.848** | 0.858 | 0.708 | 0.798 | 0.849 | **0.879** | 0.815 | 0.813 | 0.594 | **0.908** | 0.811 |
| **F1 Score** | 0.244 | 0.407 | 0.254 | 0.447 | 0.176 | 0.127 | 0.090 | 0.284 | 0.313 | 0.306 | 0.387 | 0.172 | 0.267 |
| **IoU** | 0.165 | 0.295 | 0.163 | 0.314 | 0.110 | 0.075 | 0.049 | 0.189 | 0.201 | 0.203 | 0.258 | 0.104 | 0.177 |
| **(c) Dinomaly - Binary Map - TopoOT** | | | | | | | | | | | | | |
| **Precision** | 0.398 | 0.613 | 0.459 | 0.650 | 0.490 | 0.395 | 0.363 | 0.661 | 0.649 | 0.642 | 0.738 | 0.498 | 0.546 |
| **Recall** | 0.658 | 0.553 | 0.676 | 0.648 | 0.467 | 0.569 | 0.573 | 0.505 | 0.468 | 0.458 | 0.371 | 0.695 | 0.553 |
| **F1 Score** | 0.410 | 0.497 | 0.448 | 0.584 | 0.329 | 0.432 | 0.420 | 0.532 | 0.515 | 0.501 | 0.428 | 0.470 | 0.464 |
| **IoU** | 0.175 | 0.298 | 0.177 | 0.388 | 0.129 | 0.115 | 0.097 | 0.275 | 0.268 | 0.285 | 0.329 | 0.134 | 0.223 |

Table 14 showcases a performance evaluation of three binary map strategies on the VisA dataset, with our technique, TopoOT, consistently demonstrating superior performance. Across the 12 categories, TopoOT regularly secures the highest F1 Score and Precision values. Our mean value of F1 Score **0.464** represents a substantial **+0.125** improvement over Dinomaly-Binary Map-THR ($\mu + 3\sigma$)s and **+0.197** improvement over TTT4AS. Similarly, for the average Precision, TopoOT shows an improvement of **+0.271** and **+0.323** over Dinomaly-Binary Map-THR ($\mu + 3\sigma$)s and TTT4AS, respectively. This trend of significant improvement is not limited to these instances but is a general pattern, indicating that our method also outperforms on other datasets, establishing TopoOT as a robust and highly effective technique for anomaly detection.

Table 15 presents the results of Dinomaly on Real-IAD (30 classes), with anomaly scores reported as I-AUROC, P-AUROC, and P-AUPRO. These results are reproduced directly using the official implementation provided by the authors.

Table 16 presents a performance evaluation of three binary map strategies, and our method, TopoOT, consistently demonstrates superior performance. A detailed analysis of the quantitative results reveals that

Table 15: Dinomaly (Guo et al., 2025) on Real-IAD (30 classes). I-AUROC, P-AUROC, P-AUPRO.

| Metric | audiojack | b-cap | b-battery | e-cap | eraser | f-hood | mint | mounts | pcb | p-battery | p-nut | p-plug | p-doll | regulator | r-base | s-set |
|---|---|---|---|---|---|---|---|---|---|---|---|---|---|---|---|---|
| **Dinomaly (Guo et al., 2025) — Anomaly Scores** | | | | | | | | | | | | | | | | |
| **I-AUROC** | 0.868 | 0.899 | 0.866 | 0.870 | 0.903 | 0.838 | 0.731 | 0.904 | 0.920 | 0.929 | 0.883 | 0.905 | 0.851 | 0.852 | 0.992 | 0.958 |
| **P-AUROC** | 0.917 | 0.981 | 0.929 | 0.960 | 0.964 | 0.930 | 0.776 | 0.956 | 0.957 | 0.968 | 0.974 | 0.964 | 0.960 | 0.956 | 0.985 | 0.909 |
| **P-AUPRO** | 0.917 | 0.981 | 0.929 | 0.960 | 0.964 | 0.930 | 0.776 | 0.956 | 0.957 | 0.968 | 0.974 | 0.964 | 0.960 | 0.956 | 0.985 | 0.909 |

| Metric | switch | tape | t-block | t-brush | toy | t-brick | transistor1 | u-block | usb | u-adaptor | vcpill | w-beads | woodstick | zipper | Mean |
|---|---|---|---|---|---|---|---|---|---|---|---|---|---|---|---|
| **Dinomaly (Guo et al., 2025) — Anomaly Scores** | | | | | | | | | | | | | | | |
| **I-AUROC** | 0.978 | 0.969 | 0.967 | 0.904 | 0.856 | 0.723 | 0.974 | 0.899 | 0.920 | 0.815 | 0.920 | 0.873 | 0.840 | 0.991 | 0.893 |
| **P-AUROC** | 0.959 | 0.988 | 0.988 | 0.904 | 0.910 | 0.766 | 0.978 | 0.968 | 0.975 | 0.910 | 0.937 | 0.905 | 0.904 | 0.978 | 0.989 |
| **P-AUPRO** | 0.959 | 0.988 | 0.988 | 0.904 | 0.910 | 0.766 | 0.978 | 0.968 | 0.975 | 0.910 | 0.937 | 0.905 | 0.904 | 0.978 | 0.939 |

Table 16: Performance evaluation of Dinomaly (Guo et al., 2025) across 30 classes (Real-IAD Dataset) and their mean, comparing three binary map strategies: (a) THR ($\mu + 3\sigma$), (b) TTT4AS, and (c) TopoOT. The best result for each Precision, Recall, and F1 Score is in **bold** and the second-best in blue.

| Metric | audiojack | b-cap | b-battery | e-cap | eraser | f-hood | mint | mounts | pcb | p-battery | p-nut | p-plug | p-doll | regulator | r-base | s-set |
|---|---|---|---|---|---|---|---|---|---|---|---|---|---|---|---|---|
| **(a) Dinomaly - Binary Map - THR ($\mu + 3\sigma$) (Guo et al., 2025)** | | | | | | | | | | | | | | | | |
| **Precision** | 0.366 | 0.105 | 0.274 | 0.304 | 0.164 | 0.196 | 0.144 | 0.222 | 0.383 | 0.186 | 0.159 | 0.134 | 0.193 | 0.132 | 0.170 | 0.184 |
| **Recall** | 0.645 | 0.985 | 0.435 | 0.663 | 0.832 | 0.775 | 0.664 | 0.826 | 0.719 | 0.903 | 0.885 | 0.937 | 0.737 | 0.895 | 0.996 | 0.776 |
| **F1 Score** | 0.427 | 0.186 | 0.282 | 0.350 | 0.260 | 0.290 | 0.217 | 0.325 | 0.442 | 0.299 | 0.259 | 0.229 | 0.273 | 0.215 | 0.272 | 0.279 |
| **IoU** | 0.303 | 0.105 | 0.187 | 0.234 | 0.163 | 0.183 | 0.138 | 0.217 | 0.312 | 0.185 | 0.158 | 0.133 | 0.172 | 0.130 | 0.169 | 0.183 |
| **(b) Dinomaly - Binary Map - TTT4AS (Costanzino et al., 2024a)** | | | | | | | | | | | | | | | | |
| **Precision** | 0.102 | 0.056 | 0.113 | 0.093 | 0.107 | 0.098 | 0.095 | 0.229 | 0.184 | 0.188 | 0.122 | 0.102 | 0.123 | 0.121 | 0.184 | 0.174 |
| **Recall** | 0.804 | 0.721 | 0.504 | 0.874 | 0.803 | 0.807 | 0.532 | 0.888 | 0.844 | 0.866 | 0.866 | 0.904 | 0.730 | 0.816 | 0.924 | 0.720 |
| **F1 Score** | 0.171 | 0.098 | 0.135 | 0.159 | 0.169 | 0.159 | 0.145 | 0.328 | 0.281 | 0.297 | 0.198 | 0.176 | 0.177 | 0.188 | 0.285 | 0.263 |
| **IoU** | 0.103 | 0.056 | 0.091 | 0.093 | 0.107 | 0.098 | 0.094 | 0.224 | 0.183 | 0.187 | 0.123 | 0.102 | 0.112 | 0.121 | 0.182 | 0.175 |
| **(c) Dinomaly - Binary Map - TopoOT** | | | | | | | | | | | | | | | | |
| **Precision** | 0.465 | 0.383 | 0.333 | 0.339 | 0.418 | 0.360 | 0.307 | 0.559 | 0.526 | 0.562 | 0.368 | 0.399 | 0.406 | 0.445 | 0.583 | 0.475 |
| **Recall** | 0.604 | 0.662 | 0.415 | 0.653 | 0.579 | 0.660 | 0.501 | 0.505 | 0.606 | 0.561 | 0.609 | 0.711 | 0.562 | 0.505 | 0.699 | 0.477 |
| **F1 Score** | 0.465 | 0.460 | 0.259 | 0.400 | 0.441 | 0.410 | 0.308 | 0.490 | 0.529 | 0.515 | 0.388 | 0.465 | 0.409 | 0.436 | 0.581 | 0.382 |
| **IoU** | 0.335 | 0.320 | 0.162 | 0.275 | 0.315 | 0.294 | 0.211 | 0.369 | 0.390 | 0.380 | 0.273 | 0.328 | 0.287 | 0.315 | 0.439 | 0.275 |

| Metric | switch | tape | t-block | t-brush | toy | t-brick | transistor1 | u-block | usb | u-adaptor | vcpill | w-beads | woodstick | zipper | Mean |
|---|---|---|---|---|---|---|---|---|---|---|---|---|---|---|---|
| **(a) Dinomaly - Binary Map - THR ($\mu + 3\sigma$) (Guo et al., 2025)** | | | | | | | | | | | | | | | |
| **Precision** | 0.336 | 0.190 | 0.182 | 0.427 | 0.174 | 0.310 | 0.312 | 0.190 | 0.323 | 0.094 | 0.452 | 0.310 | 0.206 | 0.452 | 0.242 |
| **Recall** | 0.931 | 0.973 | 0.967 | 0.374 | 0.647 | 0.654 | 0.884 | 0.836 | 0.874 | 0.923 | 0.740 | 0.721 | 0.831 | 0.747 | 0.793 |
| **F1 Score** | 0.467 | 0.301 | 0.296 | 0.307 | 0.219 | 0.348 | 0.438 | 0.286 | 0.431 | 0.165 | 0.499 | 0.393 | 0.310 | 0.431 | 0.317 |
| **IoU** | 0.324 | 0.189 | 0.180 | 0.205 | 0.133 | 0.236 | 0.292 | 0.187 | 0.288 | 0.094 | 0.367 | 0.271 | 0.199 | 0.293 | 0.208 |
| **(b) Dinomaly - Binary Map - TTT4AS (Costanzino et al., 2024a)** | | | | | | | | | | | | | | | |
| **Precision** | 0.154 | 0.150 | 0.160 | 0.312 | 0.109 | 0.153 | 0.200 | 0.121 | 0.159 | 0.052 | 0.296 | 0.161 | 0.154 | 0.351 | 0.154 |
| **Recall** | 0.927 | 0.928 | 0.951 | 0.553 | 0.625 | 0.860 | 0.812 | 0.794 | 0.882 | 0.702 | 0.871 | 0.838 | 0.878 | 0.815 | 0.801 |
| **F1 Score** | 0.233 | 0.240 | 0.260 | 0.342 | 0.166 | 0.236 | 0.295 | 0.192 | 0.245 | 0.087 | 0.410 | 0.250 | 0.245 | 0.430 | 0.229 |
| **IoU** | 0.148 | 0.151 | 0.159 | 0.233 | 0.105 | 0.151 | 0.187 | 0.121 | 0.155 | 0.052 | 0.292 | 0.160 | 0.153 | 0.298 | 0.147 |
| **(c) Dinomaly - Binary Map - TopoOT** | | | | | | | | | | | | | | | |
| **Precision** | 0.629 | 0.434 | 0.632 | 0.526 | 0.382 | 0.395 | 0.579 | 0.443 | 0.515 | 0.325 | 0.627 | 0.458 | 0.302 | 0.641 | 0.461 |
| **Recall** | 0.526 | 0.626 | 0.722 | 0.274 | 0.504 | 0.670 | 0.513 | 0.607 | 0.608 | 0.566 | 0.574 | 0.606 | 0.703 | 0.506 | 0.577 |
| **F1 Score** | 0.527 | 0.429 | 0.636 | 0.294 | 0.380 | 0.430 | 0.500 | 0.452 | 0.520 | 0.352 | 0.540 | 0.458 | 0.340 | 0.466 | 0.442 |
| **IoU** | 0.374 | 0.300 | 0.490 | 0.191 | 0.275 | 0.301 | 0.352 | 0.331 | 0.372 | 0.234 | 0.402 | 0.337 | 0.239 | 0.330 | 0.317 |

TopoOT frequently outperforms both Dinomaly - THR ($\mu + 3\sigma$) and Dinomaly - TTT4AS, securing the top rank for F1 Score and Precision in most categories. On average, our F1 Score of **0.442** represents a significant **+0.125** improvement over Dinomaly ($\mu + 3\sigma$)'s F1 Score of 0.0.317. Similarly, our F1 score is **+0.213** more than the Dinomaly - TTT4AS. TopoOT has a Precision of **0.461**, which is **+0.219** better than Dinomaly - THR ($\mu + 3\sigma$)'s Precision of 0.0.242 and **0.307** more than Dinomaly TTT4AS. This consistent trend of significant improvement is not limited to these instances but is a general pattern, indicating that our method also outperforms on other datasets, establishing TopoOT as a robust and highly effective technique for anomaly detection.

## A.5 ADDITIONAL QUANTITATIVE RESULTS ON 3D AD&S DATASETS

Table 17 presents the results of CMM across categories of the MVTec 3D-AD dataset, with anomaly scores reported as I-AUROC, P-AUROC, and P-AUPRO. These results are reproduced directly using the official implementation provided by the authors.

Table 17: CMM (Costanzino et al., 2024b) anomaly scores accross categories of the MVTec 3D-AD dataset (Bergmann et al., 2021).

| Metric | Bagel | Gland | Carrot | Cookie | Dowel | Foam | Peach | Potato | Rope | Tire | Mean |
|--------|-------|-------|--------|--------|-------|------|-------|--------|------|------|------|
| CMM (Costanzino et al., 2024b) – Anomaly Score | | | | | | | | | | | |
| I-AUROC | 0.994 | 0.888 | 0.984 | 0.993 | 0.980 | 0.888 | 0.941 | 0.943 | 0.980 | 0.953 | 0.954 |
| P-AUROC | 0.997 | 0.992 | 0.999 | 0.972 | 0.987 | 0.993 | 0.998 | 0.999 | 0.998 | 0.998 | 0.993 |
| P-AUPRO | 0.979 | 0.972 | 0.982 | 0.945 | 0.950 | 0.968 | 0.980 | 0.982 | 0.975 | 0.981 | 0.971 |

Table 18 reports the results of M3DM on the MVTec 3D-AD dataset, with anomaly scores given in terms of I-AUROC, P-AUROC, and P-AUPRO. The results are reproduced directly using the official implementation provided by the authors.

Table 18: M3DM (Wang et al., 2023b) anomaly scores across categories of the MVTec 3D-AD dataset (Bergmann et al., 2021).

| Metric | Bagel | Gland | Carrot | Cookie | Dowel | Foam | Peach | Potato | Rope | Tire | Mean |
|--------|-------|-------|--------|--------|-------|------|-------|--------|------|------|------|
| M3DM (Wang et al., 2023b) – Anomaly Score | | | | | | | | | | | |
| I-AUROC | 0.994 | 0.909 | 0.972 | 0.976 | 0.960 | 0.942 | 0.973 | 0.899 | 0.972 | 0.850 | 0.945 |
| P-AUROC | 0.995 | 0.993 | 0.997 | 0.985 | 0.985 | 0.984 | 0.996 | 0.994 | 0.997 | 0.996 | 0.992 |
| P-AUPRO | 0.970 | 0.971 | 0.979 | 0.950 | 0.941 | 0.932 | 0.977 | 0.971 | 0.971 | 0.975 | 0.964 |

Table 19 reports the quantitative results of our proposed method **TopoOT** against two competitive baselines, namely CMM-THR and CMM-TTT4AS, across the MVTec 3D-AD benchmark. By analysing the mean column, we observe that TopoOT consistently outperforms both baselines across multiple metrics.

In terms of Precision, TopoOT achieves a mean score of 0.427, significantly improving over CMM-THR (0.199) and CMM-TTT4AS (0.303). For Recall, TopoOT yields second best value for a mean of 0.845, and CMM–THR achieves 0.902, and CMM–TTT4AS (0.608). With respect to F1 Score, TopoOT secures a mean value of 0.482, which is a notable gain of **+0.207** compared to CMM–THR (0.275) and **+0.102** gain against CMM–TTT4AS (0.377). Similarly, for IoU, TopoOT obtains a mean of 0.343, showing clear improvements over CMM–THR (0.232) and CMM–TTT4AS (0.077).

These improvements are particularly evident in the *Gland*, *Cookie*, and *Carrot* categories, where TopoOT demonstrates substantial gains in F1 Score and IoU compared to both baseline methods. While CMM–THR exhibits high recall values, it suffers from very low precision, highlighting its bias toward over-segmentation.

Table 19: Evaluation of CMM (Costanzino et al., 2024b) across benchmarks in the MVTec 3D-AD (Bergmann et al., 2021).

| Method | Bagel | Gland | Carrot | Cookie | Dowel | Foam | Peach | Potato | Rope | Tire | Mean |
|---|---|---|---|---|---|---|---|---|---|---|---|
| **(a) CMM - THR ($\mu + 3\sigma$) (Costanzino et al., 2024b)** | | | | | | | | | | | |
| **Precision** | 0.301 | 0.188 | 0.049 | 0.518 | 0.072 | 0.275 | 0.262 | 0.092 | 0.049 | 0.182 | 0.199 |
| **Recall** | **0.949** | 0.842 | **0.998** | **0.901** | **0.896** | 0.597 | **0.957** | **0.998** | **0.989** | 0.896 | **0.902** |
| **F1 Score** | 0.425 | 0.265 | 0.092 | 0.619 | 0.129 | 0.327 | 0.375 | 0.160 | 0.091 | 0.267 | 0.275 |
| **IoU** | 0.411 | 0.182 | 0.102 | **0.578** | 0.105 | **0.276** | 0.233 | 0.085 | 0.149 | **0.198** | 0.232 |
| **(b) CMM - TTT4AS (Costanzino et al., 2024a)** | | | | | | | | | | | |
| **Precision** | 0.432 | 0.258 | 0.242 | 0.713 | 0.195 | 0.214 | 0.353 | 0.252 | 0.264 | 0.111 | 0.303 |
| **Recall** | 0.745 | 0.766 | 0.889 | 0.603 | 0.739 | 0.732 | 0.872 | 0.888 | 0.865 | 0.904 | 0.800 |
| **F1 Score** | 0.495 | 0.362 | 0.351 | 0.606 | 0.289 | 0.311 | 0.470 | 0.363 | 0.360 | 0.189 | 0.380 |
| **IoU** | 0.264 | 0.037 | 0.029 | 0.231 | 0.031 | 0.058 | 0.034 | 0.028 | 0.029 | 0.030 | 0.077 |
| **(c) CMM - TopoOT** | | | | | | | | | | | |
| **Precision** | **0.560** | **0.347** | **0.398** | **0.841** | **0.387** | **0.298** | **0.432** | **0.308** | **0.477** | **0.224** | **0.427** |
| **Recall** | 0.847 | **0.849** | 0.905 | 0.643 | 0.658 | **0.893** | 0.903 | 0.947 | 0.822 | **0.980** | 0.845 |
| **F1 Score** | **0.618** | **0.419** | **0.516** | **0.672** | **0.438** | **0.345** | **0.519** | **0.411** | **0.525** | **0.360** | **0.482** |
| **IoU** | **0.476** | **0.305** | **0.371** | 0.535 | **0.312** | 0.238 | **0.387** | **0.289** | **0.394** | 0.119 | **0.343** |

In contrast, TopoOT provides a more balanced trade-off, achieving consistently higher F1 Scores and IoU, which are more indicative of robust anomaly localisation.

Overall, the results establish that TopoOT achieves superior performance not only in terms of mean values but also across a wide range of categories, confirming its ability to generalise effectively to diverse datasets within MVTec 3D-AD.

Table 20: Evaluation of M3DM (Wang et al., 2023b) across benchmarks in the MVTec 3D-AD (Bergmann et al., 2021).

| Method | Bagel | Gland | Carrot | Cookie | Dowel | Foam | Peach | Potato | Rope | Tire | Mean |
|---|---|---|---|---|---|---|---|---|---|---|---|
| **(a) M3DM - THR ($\mu + 3\sigma$) (Wang et al., 2023b)** | | | | | | | | | | | |
| **Precision** | 0.174 | 0.105 | 0.045 | 0.493 | 0.221 | 0.254 | 0.067 | 0.050 | 0.194 | 0.127 | 0.173 |
| **Recall** | **0.949** | **0.980** | **0.997** | 0.712 | **0.909** | 0.536 | **1.000** | **0.999** | **0.917** | 0.894 | **0.889** |
| **F1 Score** | 0.270 | 0.174 | 0.085 | 0.547 | 0.328 | 0.318 | 0.121 | 0.094 | 0.308 | 0.204 | 0.245 |
| **IoU** | 0.431 | 0.189 | 0.114 | **0.552** | 0.151 | **0.333** | 0.198 | 0.117 | 0.182 | 0.053 | 0.232 |
| **(b) M3DM - TTT4AS (Costanzino et al., 2024a)** | | | | | | | | | | | |
| **Precision** | 0.498 | **0.486** | 0.337 | 0.752 | 0.464 | **0.386** | 0.536 | 0.347 | **0.561** | 0.302 | 0.467 |
| **Recall** | 0.607 | 0.706 | 0.750 | 0.351 | 0.691 | 0.624 | 0.779 | 0.684 | 0.543 | 0.669 | 0.640 |
| **F1 Score** | 0.478 | **0.525** | 0.422 | 0.443 | 0.514 | 0.440 | 0.585 | 0.419 | 0.468 | **0.383** | 0.468 |
| **IoU** | 0.287 | 0.078 | 0.031 | 0.343 | 0.066 | 0.148 | 0.090 | 0.026 | 0.099 | 0.028 | 0.120 |
| **(c) M3DM - TopoOT** | | | | | | | | | | | |
| **Precision** | **0.870** | 0.357 | **0.490** | **0.829** | **0.566** | 0.379 | **0.603** | **0.490** | 0.254 | **0.798** | **0.564** |
| **Recall** | 0.744 | 0.806 | 0.794 | 0.571 | 0.685 | **0.910** | 0.862 | 0.823 | 0.540 | **0.935** | 0.767 |
| **F1 Score** | **0.655** | 0.406 | **0.559** | **0.626** | **0.564** | **0.452** | **0.661** | **0.541** | 0.304 | 0.127 | **0.490** |
| **IoU** | **0.515** | **0.294** | **0.406** | 0.480 | **0.418** | 0.333 | **0.519** | **0.401** | 0.195 | 0.077 | **0.364** |

Table 20 presents the quantitative comparison of our proposed method **TopoOT** against two state-of-the-art baselines, M3DM–THR and M3DM–TTT4AS, across the MVTec 3D-AD benchmark. The results clearly demonstrate that TopoOT achieves consistent improvements across all metrics.

On the mean column, TopoOT achieves a Precision of 0.564, which represents an improvement of **+0.391** over M3DM–THR (0.173) and **+0.097** over M3DM–TTT4AS (0.467). In terms of Recall, our method obtains 0.767, showing a second-best result compared to M3DM–THR (0.889) and M3DM–TTT4AS (0.640). More importantly, for F1 Score, which balances precision and recall, TopoOT achieves 0.490, significantly

outperforming M3DM–THR (0.245) and M3DM–TTT4AS (0.468). Similarly, for IoU, TopoOT yields 0.364, surpassing M3DM–THR (0.232) and M3DM–TTT4AS (0.120).

Overall, the improvements in mean performance, alongside consistent category-level gains, confirm the superior generalisation ability of TopoOT across both simple and complex 3D anomaly detection scenarios in MVTec 3D-AD.

Table 21: Performance evaluation of PO3AD (Ye et al., 2025) across 29 categories of Anomaly-ShapeNet (Li et al., 2024) and their mean, comparing three binary map strategies: (a) THR ($\mu + 3\sigma$), (b) TTT4AS, and (c) TopoOT. The table highlights the best result for each Precision, Recall, and F1 Score metric in **bold black** and the second-best in blue.

| Metric | ashtray0 | bag0 | bottle0 | bottle1 | bottle3 | bowl0 | bowl1 | bowl2 | bowl3 | bowl4 | bowl5 | bucket0 | bucket1 | cap0 | cap3 |
|---|---|---|---|---|---|---|---|---|---|---|---|---|---|---|---|
| **(a) PO3AD — Binary Map — THR ($\mu + 3\sigma$) (Ye et al., 2025)** | | | | | | | | | | | | | | | |
| Precision | 0.920 | 0.678 | 0.737 | 0.714 | 0.847 | 0.797 | 0.589 | 0.815 | 0.607 | 0.872 | 0.647 | 0.709 | 0.716 | 0.781 | 0.726 |
| Recall | 0.280 | 0.362 | 0.346 | 0.326 | 0.637 | 0.301 | 0.702 | 0.639 | 0.707 | 0.746 | 0.472 | 0.256 | 0.284 | 0.275 | 0.527 |
| F1 Score | 0.417 | 0.464 | 0.460 | 0.420 | 0.720 | 0.429 | 0.630 | 0.713 | 0.644 | 0.793 | 0.539 | 0.359 | 0.387 | 0.390 | 0.720 |
| IoU | 0.272 | 0.344 | 0.331 | 0.285 | 0.586 | 0.278 | 0.482 | 0.596 | 0.496 | 0.660 | 0.410 | 0.236 | 0.263 | 0.255 | 0.487 |
| **(b) PO3AD — Binary Map — TTT4AS (Costanzino et al., 2024a)** | | | | | | | | | | | | | | | |
| Precision | 0.581 | 0.492 | 0.623 | 0.601 | 0.688 | 0.654 | 0.489 | 0.677 | 0.503 | 0.712 | 0.551 | 0.599 | 0.611 | 0.635 | 0.618 |
| Recall | 0.452 | 0.510 | 0.411 | 0.405 | 0.595 | 0.388 | 0.615 | 0.559 | 0.621 | 0.646 | 0.503 | 0.354 | 0.381 | 0.370 | 0.501 |
| F1 Score | 0.508 | 0.501 | 0.495 | 0.484 | 0.638 | 0.487 | 0.545 | 0.612 | 0.556 | 0.677 | 0.526 | 0.444 | 0.469 | 0.467 | 0.553 |
| IoU | 0.341 | 0.334 | 0.329 | 0.319 | 0.469 | 0.322 | 0.375 | 0.441 | 0.385 | 0.512 | 0.357 | 0.286 | 0.306 | 0.305 | 0.383 |
| **(c) PO3AD — Binary Map — TopoOT** | | | | | | | | | | | | | | | |
| Precision | 0.849 | 0.598 | 0.707 | 0.672 | 0.804 | 0.768 | 0.568 | 0.789 | 0.576 | 0.831 | 0.619 | 0.696 | 0.701 | 0.726 | 0.706 |
| Recall | 0.463 | 0.421 | 0.411 | 0.411 | 0.722 | 0.395 | 0.740 | 0.687 | 0.764 | 0.798 | 0.538 | 0.382 | 0.439 | 0.463 | 0.530 |
| F1 Score | 0.545 | 0.453 | 0.484 | 0.470 | 0.748 | 0.512 | 0.633 | 0.726 | 0.629 | 0.801 | 0.562 | 0.430 | 0.433 | 0.525 | 0.592 |
| IoU | 0.402 | 0.343 | 0.355 | 0.337 | 0.625 | 0.354 | 0.483 | 0.615 | 0.483 | 0.670 | 0.435 | 0.299 | 0.303 | 0.390 | 0.473 |

| Metric | cup0 | cup1 | eraser0 | headset0 | headset1 | helmet0 | helmet1 | vase1 | vase2 | vase3 | vase4 | vase7 | vase8 | vase9 | Mean |
|---|---|---|---|---|---|---|---|---|---|---|---|---|---|---|---|
| **(a) PO3AD — Binary Map — THR ($\mu + 3\sigma$) (Ye et al., 2025)** | | | | | | | | | | | | | | | |
| Precision | 0.782 | 0.524 | 0.801 | 0.649 | 0.697 | 0.239 | 0.513 | 0.404 | 0.600 | 0.572 | 0.468 | 0.627 | 0.777 | 0.777 | 0.675 |
| Recall | 0.443 | 0.326 | 0.314 | 0.339 | 0.302 | 0.215 | 0.370 | 0.336 | 0.486 | 0.292 | 0.584 | 0.733 | 0.605 | 0.572 | 0.441 |
| F1 Score | 0.558 | 0.389 | 0.436 | 0.431 | 0.411 | 0.216 | 0.411 | 0.356 | 0.520 | 0.351 | 0.503 | 0.663 | 0.663 | 0.627 | 0.500 |
| IoU | 0.401 | 0.276 | 0.301 | 0.293 | 0.269 | 0.132 | 0.276 | 0.259 | 0.383 | 0.245 | 0.383 | 0.502 | 0.562 | 0.511 | 0.371 |
| **(b) PO3AD — Binary Map — TTT4AS (Costanzino et al., 2024a)** | | | | | | | | | | | | | | | |
| Precision | 0.641 | 0.445 | 0.672 | 0.540 | 0.589 | 0.198 | 0.415 | 0.355 | 0.511 | 0.498 | 0.417 | 0.533 | 0.655 | 0.661 | 0.562 |
| Recall | 0.512 | 0.455 | 0.389 | 0.458 | 0.399 | 0.311 | 0.544 | 0.410 | 0.501 | 0.321 | 0.540 | 0.588 | 0.619 | 0.582 | 0.485 |
| F1 Score | 0.569 | 0.450 | 0.493 | 0.496 | 0.476 | 0.242 | 0.471 | 0.381 | 0.506 | 0.390 | 0.470 | 0.559 | 0.637 | 0.619 | 0.510 |
| IoU | 0.398 | 0.290 | 0.327 | 0.329 | 0.312 | 0.138 | 0.308 | 0.235 | 0.339 | 0.242 | 0.307 | 0.388 | 0.467 | 0.448 | 0.347 |
| **(c) PO3AD — Binary Map — TopoOT** | | | | | | | | | | | | | | | |
| Precision | 0.746 | 0.446 | 0.783 | 0.571 | 0.666 | 0.156 | 0.318 | 0.364 | 0.548 | 0.566 | 0.432 | 0.603 | 0.745 | 0.733 | 0.631 |
| Recall | 0.549 | 0.571 | 0.368 | 0.543 | 0.370 | 0.444 | 0.666 | 0.460 | 0.523 | 0.349 | 0.622 | 0.646 | 0.723 | 0.659 | 0.540 |
| F1 Score | 0.613 | 0.426 | 0.478 | 0.486 | 0.449 | 0.223 | 0.388 | 0.360 | 0.518 | 0.387 | 0.489 | 0.611 | 0.697 | 0.666 | 0.529 |
| IoU | 0.468 | 0.310 | 0.342 | 0.353 | 0.300 | 0.135 | 0.259 | 0.255 | 0.377 | 0.278 | 0.371 | 0.465 | 0.612 | 0.559 | 0.402 |

As shown in Table 21, our method TopoOT consistently outperforms THR and TTT4AS across all metrics on Anomaly-ShapeNet. In the mean column, TopoOT achieves notable gains, +0.099 in Recall, +0.029 in F1 Score, and +0.031 in IoU over THR, and even larger improvements over TTT4AS (e.g., +0.069 in Precision, +0.055 in Recall, and 0.019 in F1 Score). These results, along with strong performance across individual categories, demonstrate that TopoOT not only sets a new state of the art but also generalises robustly across diverse anomaly types and datasets.

Table 22 reports the results of PO3AD, with anomaly scores evaluated using Object-AUROC, Point-AUROC, and Object-AUCPR. The results are reproduced directly using the official implementation provided by the authors.

Table 22: PO3AD (Ye et al., 2025) — Anomaly scores, Object-AUROC, Point-AUROC, Object-AUCPR.

| Metric | ashtray0 | bag0 | bottle0 | bottle1 | bottle3 | bowl0 | bowl1 | bowl2 | bowl3 | bowl4 | bowl5 | bucket0 | bucket1 | cap0 | cap3 |
|---|---|---|---|---|---|---|---|---|---|---|---|---|---|---|---|
| | PO3AD (Ye et al., 2025) — Anomaly Scores | | | | | | | | | | | | | | |
| **O-AUROC** | 1.000 | 0.833 | 0.900 | 0.933 | 0.926 | 0.922 | 0.829 | 0.833 | 0.881 | 0.981 | 0.849 | 0.853 | 0.787 | 0.877 | 0.859 |
| **P-AUROC** | 0.962 | 0.949 | 0.912 | 0.844 | 0.880 | 0.978 | 0.914 | 0.918 | 0.935 | 0.967 | 0.941 | 0.755 | 0.899 | 0.957 | 0.948 |
| **O-AUCPR** | 0.999 | 0.809 | 0.927 | 0.959 | 0.962 | 0.946 | 0.905 | 0.888 | 0.927 | 0.985 | 0.904 | 0.923 | 0.882 | 0.841 | 0.906 |

| Metric | cup0 | cup1 | eraser0 | headset0 | headset1 | helmet0 | helmet1 | vase1 | vase2 | vase3 | vase4 | vase7 | vase8 | vase9 | Mean |
|---|---|---|---|---|---|---|---|---|---|---|---|---|---|---|---|
| | PO3AD (Ye et al., 2025) — Anomaly Scores | | | | | | | | | | | | | | |
| **O-AUROC** | 0.871 | 0.833 | 0.995 | 0.808 | 0.923 | 0.762 | 0.961 | 0.742 | 0.952 | 0.821 | 0.675 | 0.966 | 0.739 | 0.830 | 0.867 |
| **P-AUROC** | 0.909 | 0.932 | 0.974 | 0.823 | 0.907 | 0.878 | 0.948 | 0.882 | 0.978 | 0.884 | 0.902 | 0.982 | 0.950 | 0.952 | 0.919 |
| **O-AUCPR** | 0.879 | 0.870 | 0.995 | 0.765 | 0.914 | 0.864 | 0.961 | 0.789 | 0.963 | 0.902 | 0.824 | 0.971 | 0.833 | 0.904 | 0.903 |

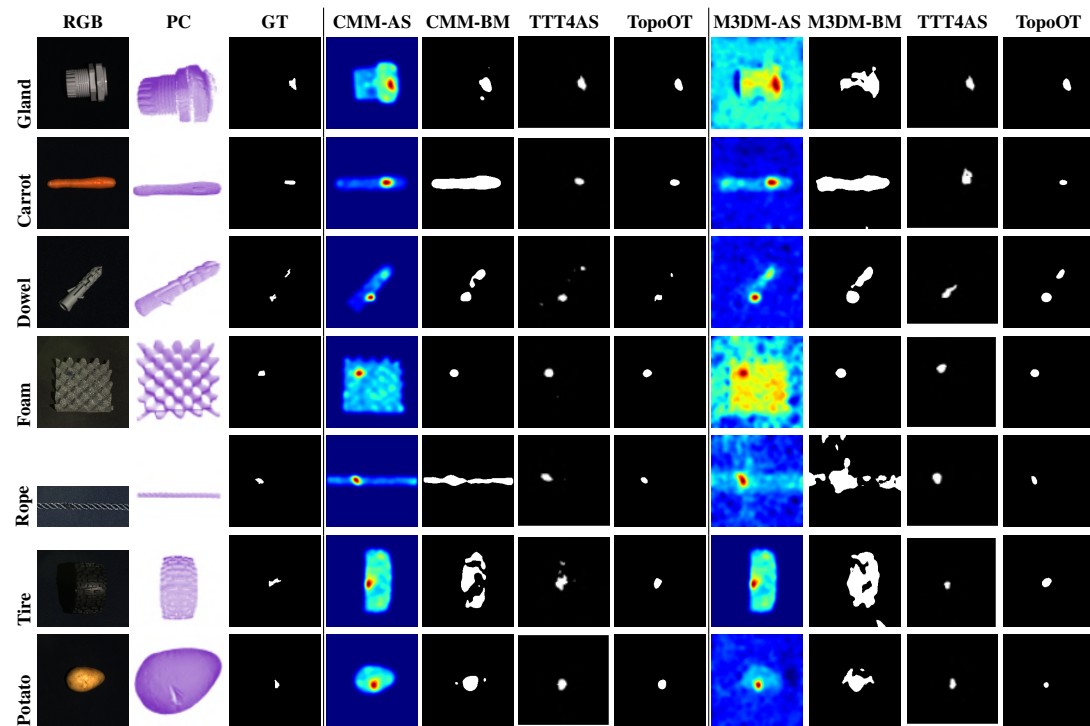

Figure 4: Qualitative comparison of AD&S methods for different objects using on 3D MvTec AD Dataset.

## A.6 OPTIMAL TRANSPORT PRELIMINARIES

For completeness, we recall the Optimal Transport (OT) formulations underlying Eq. equation 2. Let $P = \{p_i, w_i\}_{i=1}^m$ and $Q = \{q_j, v_j\}_{j=1}^n$ be two discrete probability measures with weights $w \in \Delta^m$, $v \in \Delta^n$, and cost matrix $C(i, j) = \|p_i - q_j\|_2^2$. The classical 2-Wasserstein distance is defined as

$$W_2^2(P, Q) \;=\; \min_{\Pi \in \mathcal{U}(w,v)} \langle C, \Pi \rangle,$$

where $\Pi \in \mathbb{R}_+^{m \times n}$ is a transport plan and $\mathcal{U}(w, v) = \{\Pi \mid \Pi \mathbf{1} = w, \ \Pi^\top \mathbf{1} = v\}$ denotes the set of admissible couplings. While exact OT provides a principled alignment, solving this linear program has $O(m^3 \log m)$ complexity, and the resulting optimal plans are typically sparse. In practice, sparsity can make OT couplings numerically sensitive, that is, small perturbations in the support points may lead to abrupt changes in the optimal plan (Peyré et al., 2019).

To improve robustness and computational efficiency, we adopt the *entropy-regularised* variant, known as the Sinkhorn distance (Cuturi, 2013; Peyré et al., 2019):

$$W_\varepsilon(P, Q) \;=\; \min_{\Pi \in \mathcal{U}(w,v)} \langle C, \Pi \rangle \;+\; \varepsilon H(\Pi),$$

where $H(\Pi) = \sum_{i,j} \Pi(i, j)(\log \Pi(i, j) - 1)$ is the negative entropy of $\Pi$. The regularisation parameter $\varepsilon > 0$ controls smoothness: large $\varepsilon$ yields dense couplings, while small $\varepsilon$ approaches the exact Wasserstein distance.

In our pipeline, persistence diagrams are constructed using `GUDHI` (cubical complexes), but all transport computations are carried out with `POT`'s `ot.sinkhorn(..., reg=`$\varepsilon$`)` routine[2]. Thus, the couplings $\Pi^\star$ appearing in Sec. 3.2 and Appendix A.7 are entropy-regularised OT plans. This choice ensures numerical stability, differentiability, and Lipschitz continuity, which underlie the stability and generalisation guarantees established in Appendix A.7.

## A.7 CONCEPTUAL MOTIVATION

A central motivation of our framework is that anomaly segmentation under distribution shift can be interpreted through the discrepancy between distributions of persistence features. Let $\mathcal{D}_{\text{sub}}$ and $\mathcal{D}_{\text{sup}}$ denote the empirical distributions of birth–death components extracted from the sub- and super-level filtrations (Sec. 3.1). The entropic OT distance

$$W_\varepsilon(\mathcal{D}_{\text{sub}}, \mathcal{D}_{\text{sup}}) = \min_{\Pi \in \mathcal{U}(\mathcal{D}_{\text{sub}}, \mathcal{D}_{\text{sup}})} \langle C, \Pi \rangle + \varepsilon H(\Pi)$$

quantifies the minimal cost of aligning structural information across the two filtrations. Computing $W_\varepsilon$ identifies components with stable, low-cost couplings, from which OT-guided pseudo-labels $\widetilde{Y}_{\text{OT}}$ are derived (Sec. 3.2). By combining the classical stability of persistence diagrams with the smooth dependence of entropic OT on point locations, this construction is expected to yield pseudo-labels that are more stable under small local perturbations of the anomaly map.

Beyond stability, this perspective connects conceptually to classical discrepancy-based domain adaptation (DA). In the DA setting (Redko et al., 2017), the target risk can be upper bounded by a source risk plus a discrepancy term (e.g., a Wasserstein distance). We use this framework purely as an analogy: in our setting, the "source" and "target" distributions correspond to persistence features extracted at different filtration levels or under distribution shift. We do not train hypotheses within the OT step, nor do we claim a new DA

---

[2]https://pythonot.github.io/

bound; the analogy simply clarifies why reducing OT discrepancy across filtrations correlates with empirical robustness.

**Setup.** Let $P_k^f$ denote the persistence diagram extracted from the $f \in \{\text{sub}, \text{sup}\}$ filtration at threshold $\tau_k$. We compute entropic OT distances between augmented diagrams (Sec. A.6), allowing each point $p = (b, d)$ to match either a point in another diagram or its diagonal projection. Let $\Pi_{k\to\ell}^{\star}$ be the optimal transport plan between $P_k^f$ and $P_\ell^g$ with ground cost $C(i, j) = \|p_i - q_j\|_2^2$. The cross-level stability score $s(c)$ for a feature $c$ is defined in Sec. 3.2.

**Observation: Stability under perturbations.** Persistence diagrams are stable under perturbations of the underlying function, in the sense that moving each point by at most $\rho$ perturbs the diagram by at most $O(\rho)$ in standard diagram distances. Entropically regularised OT inherits this smooth dependence on point positions. Consequently, the chained stability scores used for feature selection vary smoothly under $\rho$-bounded perturbations. Features separated by a sufficiently large margin retain their relative ranking.

**Observation: Behaviour of entropic OT along chains.** The entropic OT $W_\varepsilon$ debiased counterpart, the Sinkhorn divergence $S_\varepsilon$, is a true metric and obeys a triangle inequality (Feydy et al., 2019). This provides a useful analogy for interpreting chained OT behaviour, if pairwise discrepancies along a filtration chain decrease, the corresponding Sinkhorn divergence between the endpoints also decreases. Although our method operates directly on $W_\varepsilon$, we observe empirically that reducing local OT costs across levels suppresses spurious cross-level inconsistencies, consistent with the behaviour suggested by the metric structure of $S_\varepsilon$.

**Interpretation.** Together, the stability of persistence diagrams and the behaviour of entropic OT provide intuition for why the OT-chaining mechanism is robust and why it can reduce cross-level discrepancy in practice. These results are conceptual and do not constitute a new formal theory; they serve to situate the empirical behaviour observed in our experiments within existing stability principles from topological data analysis and discrepancy-based generalisation theory.

## A.8 CUBICAL PERSISTENCE

A *primitive interval* is $J = [k, k+1] \subset \mathbb{R}$ with $k \in \mathbb{Z}$, called a 1-cube, the degenerate case $[k]$ is a 0-cube. A $d$-dimensional *elementary cube* is the Cartesian product

$$CU = J_1 \times \cdots \times J_d \in \mathbb{R}^d, \tag{5}$$

e.g., vertices, edges, squares, and voxels in 3D.

The boundary of $CU$ is

$$\partial CU = \sum_{i=1}^{d} (-1)^{i+1} (J_1 \times \cdots \times \partial J_i \times \cdots \times J_d), \tag{6}$$

where $\partial J_i = \{k, k+1\}$. A cube $CU$ is a *subcube* of $CU'$ if $J_i \subseteq J_i'$ for all $i$.

A *cubical complex* $\mathcal{K}$ is a set of cubes closed under subcubes and boundaries, ensuring structural coherence across dimensions (Fig. 5).

The chain group $CU_n(K)$ is the free Abelian group on $n$-cubes, linked by boundary maps

$$\cdots \to CU_{n+1}(K) \xrightarrow{\partial_{n+1}} CU_n(K) \xrightarrow{\partial_n} CU_{n-1}(K) \to \cdots,$$

with $\partial_n \circ \partial_{n+1} = 0$. Cycles and boundaries are

$$Z_n(K) = \ker(\partial_n), \quad B_n(K) = \text{im}(\partial_{n+1}),$$

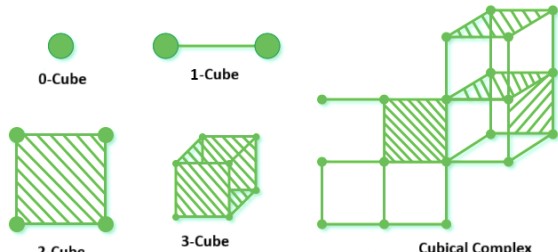

Figure 5: Elementary cubes of different dimensions and an example cubical complex.

and the $n$-th homology group is $H_n(K) = Z_n(K)/B_n(K)$.

A *filtration function* $f_K : K \to \mathbb{R}$ activates cubes monotonically: $P \sqsubseteq Q \Rightarrow f_K(P) \leq f_K(Q)$. This defines sublevel and superlevel sets:

$$K(a_i) = f_K^{-1}((-\infty, a_i]), \quad K^{\uparrow}(b_i) = f_K^{-1}([b_i, +\infty)). \tag{7}$$

Filtrations induce homology maps

$$H_k(K_0) \xrightarrow{\varphi_{01}} H_k(K_1) \xrightarrow{\varphi_{12}} \cdots \xrightarrow{\varphi_{n-1,n}} H_k(K_n),$$

forming the persistence module

$$\mathcal{P} = \{H_k(K_i), \varphi_{ij}\}_{0 \leq i \leq j \leq n}.$$

Each topological feature $\sigma$ has birth $b_\sigma$, death $d_\sigma$, and persistence $d_\sigma - b_\sigma$. The collection of intervals $[b_\sigma, d_\sigma)$ forms the *barcode*, while the *persistence diagram* (PD) encodes these as birth–death points in $\mathbb{R}^2$. To integrate with ML models, PDs are vectorised via

$$\Phi : \mathrm{PD} \to \mathbb{R}^M.$$

### A.9 QUALITATIVE ANALYSIS ON TEXTURAL ANOMALY CASES

Figure 6 shows typical challenging cases for TopoOT on texture-heavy categories (carpet, grid, wood). The first columns display the RGB image, the backbone anomaly heatmap, and the ground-truth (GT) mask. The next two columns show the binary segmentations obtained from the sublevel and superlevel filtrations of the anomaly score map, followed by the OT-guided pseudo-label produced by cross-filtration alignment. The last two columns compare the final TopoOT output with the TTT4AS baseline.

In these examples, the backbone anomaly scores already exhibit diffuse, fine-grained patterns with weak topological structure. The sublevel and superlevel filtrations therefore produce several small, fragmented components that only partially cover the true anomalous region, and sometimes include spurious islands in normal areas. OT chaining removes part of this noise and focuses the support, and the final TopoOT masks remain visually cleaner and closer to the GT than TTT4AS. However, the results are still not perfect: some anomalies are under-segmented, and small false positives remain, reflecting the limited topological signal in the underlying anomaly map. These cases highlights the limitations of our current design. First, TopoOT is fundamentally constrained by the quality of the backbone anomaly scores. The backbone does not produce a clear topological contrast between normal and anomalous texture, our cubical filtrations and persistence diagrams cannot recover it. Second, the test-time training head is fully unsupervised, so we have no additional supervision or explicit shape priors to correct these subtle texture errors. This suggests two natural directions for improvement: designing richer filtration functions (for example, combining anomaly

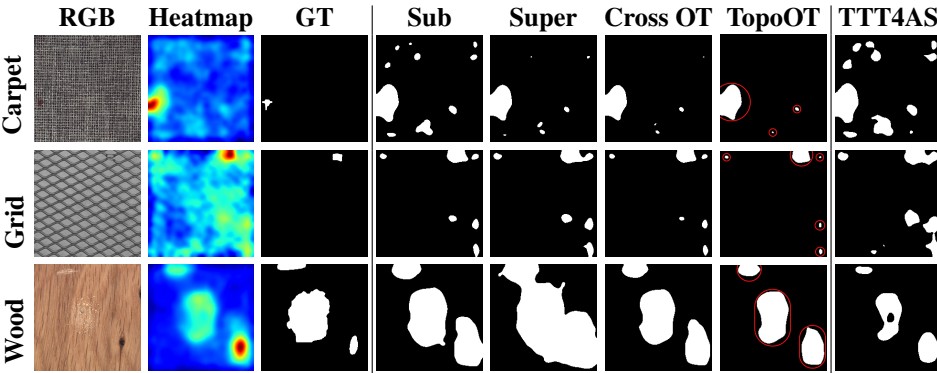

Figure 6: Qualitative examples on texture-heavy categories (carpet, grid, wood). From left to right: RGB image, backbone anomaly heatmap, ground truth (GT), binary masks from sublevel and superlevel filtrations, OT-guided pseudo-label (Cross OT), final TopoOT prediction, and TTT4AS. TopoOT reduces spurious fragments and sharpens the region compared to TTT4AS, but residual under-segmentation and small false positives remain, illustrating the challenges posed by weak topological signal in purely textural anomalies.

scores with local texture statistics or multi-scale smoothing to strengthen topological cues), and augmenting the loss with additional regularisation terms that penalise overly fragmented or isolated pseudo-labels. Even in these difficult textural regimes, the qualitative examples indicate that our topology-guided masks improve over TTT4AS while making the challenging modes interpretable in terms of the anomaly-score geometry.

## A.10 ABLATION STUDY ON TOP-K COMPONENTS

We performed a dedicated sensitivity analysis of the Top-K selection. As shown in the Table 23, we evaluate $K \in 1, 2, 3, 4, 5$ across datasets and backbones. We find that a fixed value of K=1 yields the most stable and highest F1-scores (PatchCore: 0.522; CMM: 0.482; M3DM: 0.490). As K increases, F1 consistently decreases, even though recall rises. This trend is expected; the highest-ranked components are those with the strongest OT-stability and largest persistence, whereas lower-ranked components correspond to short-persistence, less reliable structures. Including these additional components introduces noise into the pseudo-labels and degrades precision, leading to lower F1. Since K=1 is the most robust choice across datasets and architectures, we fix it globally in all experiments.

Table 23: Effect of retaining the Top-$K$ OT-stable components on anomaly segmentation. Each row corresponds to keeping the $K$ highest-ranked components (ranked by OT-stability and persistence).

| Top-$K$ Components Retained | 2D-PatchCore | | | 3D-CMM | | | 3D-M3DM | | |
|---|---|---|---|---|---|---|---|---|---|
| | Precision | Recall | F1 | Precision | Recall | F1 | Precision | Recall | F1 |
| K = 1 | **0.550** | 0.720 | **0.522** | **0.427** | **0.845** | **0.482** | **0.564** | 0.767 | **0.490** |
| K = 2 | 0.462 | 0.818 | 0.474 | 0.411 | 0.753 | 0.410 | 0.323 | 0.809 | 0.434 |
| K = 3 | 0.405 | 0.829 | 0.431 | 0.392 | 0.671 | 0.403 | 0.286 | 0.950 | 0.356 |
| K = 4 | 0.358 | 0.901 | 0.415 | 0.381 | 0.666 | 0.397 | 0.177 | 0.961 | 0.334 |
| K = 5 | 0.325 | **0.911** | 0.380 | 0.354 | 0.576 | 0.388 | 0.121 | **0.966** | 0.199 |

## A.11 Detailed TopoOT Algorithm

---

**Algorithm 1:** TopoOT: Topology-Aware Optimal Transport for Anomaly Segmentation

---

**Input** : Test image $x$; Frozen Backbone $F(\cdot)$; Thresholds $\mathcal{T} = \{\tau_1 < \cdots < \tau_N\}$; OT Reg. $\varepsilon$; Weights $\alpha, \lambda, m$; Top-$K$.

**Output :** Binary Segmentation Mask $\hat{Y}^{\text{bin}}$

1 Extract backbone features $Z = F(x)$ and scalar anomaly map $A(x) \in [0,1]^{H \times W}$;

/* **Multi-Scale Filtering (Sec. 3.1)** */

2 **foreach** *filtration type* $f \in \{sub, sup\}$ **do**

3    Initialize empty diagram list $\mathcal{D}_f \leftarrow [\,]$;

4    **foreach** *threshold* $\tau_k \in \mathcal{T}$ **do**

5       Construct cubical complex $K_{\tau_k}^f$ on the level set $S_{\tau_k}^f$ (as defined in Sec.3.1);

6       Compute persistence diagrams $P_h^f[\tau_k] = \text{PH}_h(K_{\tau_k}^f)$ for $h \in \{0,1\}$;

7       Append $\{P_h^f[\tau_k]\}_{h \in \{0,1\}}$ to $\mathcal{D}_f$;

/* **Stability Scoring (Sec. 3.2)** */

8 **foreach** $f \in \{\text{sub}, \text{sup}\}$ **do**

9    Initialize feature chains $\mathcal{C}_f$ from $\mathcal{D}_f$;

10    **foreach** *sequential pair* $(P_k, P_{k+1})$ *in* $\mathcal{D}_f$ **do**

11       Compute Cost Matrix $C$; Solve Entropic OT $\Pi_{intra}^*$ ;       // Eq. 1

12       **foreach** *feature* $c$ *in chain* **do**

13          $s_{intra}(c) \leftarrow \max_j \left( \frac{\Pi_{intra}^*(i(c),j)}{1+\sqrt{C(i(c),j)}} \right) \cdot \alpha \cdot \text{pers}(c)$ ;     // Eq. 2

14    Filter $\mathcal{C}_f$: Retain chains with high cumulative $s_{intra}$;

15 Compute OT plan $\Pi_{cross}^*$ between surviving sets $\mathcal{C}_{\text{sub}}$ and $\mathcal{C}_{\text{sup}}$;

16 **foreach** *candidate* $c \in \mathcal{C}_{\text{sub}} \cup \mathcal{C}_{\text{sup}}$ **do**

17    $s_{cross}(c) \leftarrow \max_j \left( \frac{\Pi_{cross}^*(i(c),j)}{1+\sqrt{C(i(c),j)}} \right) \cdot \alpha \cdot \text{pers}(c)$;

18 $\mathcal{C}^* \leftarrow$ Select Top-$K$ ranked candidates based on $s_{cross}(c)$ ;     // See Ablation A.10

/* **Backprojection to Pixel Space (Sec. 3.2)** */

19 Initialize pseudo-label mask $\tilde{Y}_{\text{OT}} \leftarrow 0$ on $\Omega$;

20 **foreach** *candidate* $c \in \mathcal{C}^\star$ **do**

21    Retrieve death time $d_c$ of $c$ from its persistence diagram;

22    Set backprojection threshold $\tau_{\text{bp}}(c) \leftarrow d_c$ Define pixel support $\Gamma(c) \leftarrow \{ p \in \Omega : A(p) \geq \tau_{\text{bp}}(c) \}$;

23    Update mask $\tilde{Y}_{\text{OT}}(p) \leftarrow \tilde{Y}_{\text{OT}}(p) \vee \mathbf{1}_{\Gamma(c)}(p)$ for all $p$;

/* **TopoOT Test-Time Training (Sec. 3.3)** */

24 Initialize lightweight head $h_\psi$ (MLP);

25 **while** *not converged* **do**

26    Forward: $\hat{Y}_{logits} = h_\psi(Z)$, $\hat{Y}_{prob} = \sigma(\hat{Y}_{logits})$, $z_p = \text{Normalize}(\hat{Y}_{logits}[p])$;

27    $\mathcal{L}_{OT} = \|\hat{Y}_{prob} - \tilde{Y}_{OT}\|_2$;

28    Sample pixel pairs $(p, q)$ based on $\tilde{Y}_{OT}$ (Same/Diff class);

29    $\mathcal{L}_{con} = (1 - y_{pq})\|z_p - z_q\|_2^2 + y_{pq}[\max(0, m - \|z_p - z_q\|_2)]^2$ ;     // Eq. 3

30    Update $\psi \leftarrow \psi - \eta \nabla_\psi (\mathcal{L}_{OT} + \lambda \mathcal{L}_{con})$;

/* **Inference** */

31 **return** $\hat{Y}^{\text{bin}} \leftarrow \text{AdaptiveDecisionRule}(h_\psi(Z))$;

---

