# OpenReview forum: "TopoOT: Topology-Aware Optimal Transport for Test-Time Anomaly Segmentation"
_ICLR.cc/2026/Conference — ICLR 2026 Conference Desk Rejected Submission_

### Official Review · Reviewer_sSap · 2025-10-27

**Soundness:** 2
**Presentation:** 2
**Contribution:** 2
**Rating:** 2
**Confidence:** 3

**Summary:**

This paper proposes a new method, based on optimal transport and persistent diagrams, for anomaly detection. Their key idea is that, by encoding samples through persistent diagrams, the authors are able to define anomaly maps that are topology aware. Based on this view of the data, the authors use optimal transport theory to produce an anomaly score based on their cross-scale consistency.

**Strengths:**

__Originality.__ To the best of my knowledge the authors' approach is a new combination of ideas in optimal transport and persistent diagrams for OT. I also point out that the combination of these ideas is well motivated in the main paper.

__Clarity.__ The paper is mostly clear, though the discussion in Section 3 is too high level. I comment more on this point in the weaknesses.

__Significance.__ Proposing a method that can perform anomaly detection without relying on the ad-hoc setting of a threshold, and that is robust under distributional shift is significant. With that said, the empirical validation falls a bit short of distributional shift claim (see weaknesses). I also think that the theoretical analysis would make the paper much stronger, but, as is, they are only shown in the appendix and are preliminary in nature.

**Weaknesses:**

__Clarity issues.__ Overall, I think the discussion in section 3 is too high level. Here is a list of issues,

- The formal definition of persistent diagrams is never really provided, and the overall background on these concepts is not provided in the main paper.
- The notation used in eq. 2, $i(c)$ is never really defined.
- The pseudo-label generation mechanism is mentioned, but never defined with an explicit equation or algorithm
- There is no algorithm summarizing the test-time training strategy.

__Empirical Validation.__ I like the overall structure of the experiments, especially the inclusion of a cross-domain setting that helps testing the claim that their method is robust to distributional shift. However, Table 2 only includes comparisons between cross-model experiments. I think it would be nice to have the same comparison for other TTT methods.

__Theoretical Results.__ Even though the theoretical results are not included in the main paper, they are still part of the supplementary results, and the authors mention them in the main paper. I generally don't think it is a good idea to mention preliminary results in a submission -- either the authors provide solid, complete results, they don't provide them at all, or they wait and submit once the analysis is complete. With that in mind, here are some issues with the theoretical results section in the appendix,

- The 4th equation in page 29 (the bound on the target risk by the source risk + the Wasserstein distance) was first established in (Redko et al., 2017), so the authors should cite the correct source here.
- Still on that equation, the authors never define, mathematically, $\epsilon_{S}(h)$. I think they should take the time to adapt the concepts in (Ben-David et al., 2010) and (Redko et al., 2017) to the anomaly detection scenario, which is far from obvious.
- Similarly to the main paper, the proofs presented in this section are too high level to be considered. While some of the reasoning is intuitive, the authors would need to work out the proofs rather than providing sketches.

# References

(Redko et al., 2017) Redko, Ievgen, Amaury Habrard, and Marc Sebban. "Theoretical analysis of domain adaptation with optimal transport." Joint European Conference on Machine Learning and Knowledge Discovery in Databases. Cham: Springer International Publishing, 2017.

**Questions:**

Overall, I think the authors should rework the methodological and theoretical explanations of their paper. See for instantance __Clarity issues__ and __Theoretical Results.__ under weaknesses.

---

> ### Author Response · Authors · 2025-11-20
> **Response to reviewer sSap**
>
> We are encouraged that the reviewer found the approach novel, the motivation well grounded, and the paper mostly clear. The significance of a threshold-free, topology-aware anomaly detection method, especially under distributional shift (cross-domain), was also acknowledged. We further appreciate the positive remarks on our experimental design, including the cross-domain robustness evaluation. Below, we address the reviewer's concerns.
>
>
> **Question 1:**
> Clarity issues.
>
> **Response:**
> 1. *Persistence Diagrams:* Section 3.1 (lines 186 to 198 of the original text) introduces the filtrations, birth–death pairs. References such as Zia et al. (24) (Topological Deep Learning: A Review of an Emerging Paradigm, Artificial Intelligence Review), already cited in the introduction and section 3.1, provide a detailed background on PDs, and Appendix A.8 provides the full formal background on cubical complexes and persistent homology. The detailed definitions remain available in the cited references and the appendix.
>
> 2. *Notation i(c):* In Eq. (2), *i(c)* denotes the index of the birth–death pair corresponding to candidate feature *c* within its persistence diagram. *c* is first defined in section 3.1 at the start of paragraph 3 (line 194 of original text). Subsequently, *i* and *j* are defined in lines 235 and 236 and further elaborated in line 239 in the original paper. We have added more explicit clarification of *i(c)* near Eq. (2) in the revision.
>
> 3. *Pseudo-label Generation:* The pseudo-label generation mechanism is formally defined in Sec. 3.1–3.2. Equations (1) and (2) define the stability score used to select the components. The final paragraph of Sec. 3.2 describes how the retained candidates form the set $C^\star$, which is then projected to pixel space to obtain the pseudo-labels $\tilde{Y}_{\mathrm{OT}}$.
>
> 4. *Algorithm:* The full test-time training (TTT) process is described in Sec. 3.3. We agree that presenting these steps in a compact algorithmic form would further improve readability. In the final version, we will include a concise algorithm box that summarises the TTT update procedure step-by-step.
>
> ---
>
> **Question 2:**
> Empirical Validation.
>
> **Response:**
> We appreciate that you found the overall structure of the experiments, and in particular, the cross-domain setup, effective for testing robustness under distribution shift. This cross-domain evaluation is an important part of our design, and we are glad it resonated with you.
>
> Regarding anomaly-segmentation test-time training (TTT), the only prior method we are aware of is TTT4AS. We already provide a full, same-domain comparison against TTT4AS (and the corresponding backbone anomaly maps) in the main experimental tables, where TopoOT consistently improves over this baseline.
>
> Table 2, however, serves a different purpose: it is a cross-domain **stress test** of our own architecture, where we intentionally change both the backbone and the dataset to evaluate robustness far beyond the regime considered in prior TTT work. The TTT4AS paper does not define or report any cross-domain protocol for anomaly segmentation, and no other TTT methods exist for this task.
>
> Furthermore, please see our response to question 1 of the reviewer NVIX, which defines the evaluation protocol.
>
> ---
>
> **Question 3:**
> Theoretical Results.
>
> **Response:**
> We would like to clarify that the material in Appendix A.7 **was never intended as a formal theoretical contribution.** In the appendix, it is explicitly framed as **guiding intuition** for our **hypothesis**, included only to provide additional perspective on how OT-based stability relates to the behaviour observed in the experiments. None of the claims or conclusions in the main paper depends on this subsection. The primary contribution of the work is **empirical**. As acknowledged in the overall discussion, the method is validated in a plug-and-play manner across 7 different backbones and 5 heterogeneous datasets spanning 2D, 3D, and point-cloud modalities. The ablation studies quantitatively demonstrate the robustness and generality of the approach across architectures and domains. We will revise Appendix A.7 to ensure the intention is clear, including correcting the attribution of the DA-style inequality and clarifying notation. If needed, the subsection can be further shortened, as its role is purely explanatory and does not affect the empirical findings.

---

> > ### Comment · Reviewer_sSap · 2025-11-25
> > **Response to the Authors**
> >
> > Dear authors,
> >
> > Thank you for your rebuttal. Here are a few answers to the points you mentioned,
> >
> > **Clarity**
> >
> > 1) The **main paper** should be self contained, in the sense that all definitions and mathematical equations necessary for the technical understanding of the paper should be present in the main text. Relying on external references or the appendix for core definitions may make the paper difficult to approach for non-specialist audiences. In that sense, at least the mathematical definition of the core objects treated in this paper are expected to be present in Section 3.
> >
> > 2) Thank you for adding the explicit mention. The equation is clearer now.
> >
> > 3) As I initially mentioned in my rebuttal, the authors do not provide any equation or algorithm explicitly defining how they obtain the pseudo-labels. They mention in the main paper "The surviving candidates in $\mathcal{C}^{\star}$ are projected back to their pixel-level supports on the anomaly map, yielding OT-guided pseudo-labels $\tilde{Y}\_{OT}$". **This is far from clear** and it is a too high level description of what the authors are doing.
> >
> > 4) I can't really re-evaluate this point as, so far, the authors did not provide the algorithm summarizing their approach.
> >
> > **Empirical validation**
> >
> > I consider that my questions about the empirical validation to be properly addressed.
> >
> > **Theoretical Results**
> >
> > I respectfully disagree with the authors, when they say that the "theoretical results" in their appendix were framed as "guiding intuition". The name of the section is "Theoretical Insights" and the "results" are framed as formal mathematical results (Lemmas, Propositions and Theorems) with proof sketches. As I mentioned in my initial review, "either the authors provide solid, complete results, they don't provide them at all, or they wait and submit once the analysis is complete".
> >
> > To be honest, I do think the formulation of the problem under the DA formalism is promising, but I think the authors should work more on these ideas before the paper is accepted.
> >
> > **Conclusion**
> >
> > While I think this paper is interesting and the empirical validation is solid, I still think this paper has structural problems in its presentation, and that the presentation of the "theoretical insights" in the appendix is troubling. I am inclined to keep my score at this point.

---

> > > ### Author Response · Authors · 2025-11-27
> > > **Second response to the reviewer  sSap**
> > >
> > > We appreciate the reviewer’s continued engagement and the clarifications offered. The manuscript has been updated accordingly, and we provide a response to the second-round comments below:
> > >
> > > **Clarity**
> > >
> > > 1.  In the revised manuscript, Sections 3.1 and 3.2 now include a concise mathematical definition of persistence diagrams and other constructs. These additions ensure the section is better self-contained.
> > >
> > > 2.  Thanks for your positive comment.
> > >
> > > 3. In the revised manuscript, Section 3.2 , we provide a complete definition of the backprojection step.
> > >
> > > 4. Following the reviewer’s request, we have added the complete algorithm in Appendix A.11. We also note that the full implementation (code) is provided with the submission.
> > >
> > > **Empirical validation**
> > >
> > > We appreciate the reviewer’s confirmation that the concerns regarding empirical validation have been fully addressed.
> > >
> > > **Theoretical Results**
> > >
> > > We appreciate the reviewer’s clarification. We agree that Appendix A.7, in its previous form, was not a complete theoretical development. It was only meant to offer intuition and is not used in any of the algorithms, results, or conclusions in the main paper, so we have revised it to make this limited role explicit and to avoid giving the impression of a formal theoretical contribution. We also made this explicit in the main paper (Section 3.3). The updated version is shorter and clearly presented as optional context rather than part of the core methodology.
> > >
> > > We believe these changes will address the structural issues raised in the review while preserving the contributions that the reviewer found interesting and empirically solid.

---

### Official Review · Reviewer_HFSF · 2025-10-30

**Soundness:** 2
**Presentation:** 2
**Contribution:** 2
**Rating:** 4
**Confidence:** 3

**Summary:**

This paper tackles anomaly segmentation under distribution shift and proposes TopoOT, a test-time training framework integrating persistent homology and optimal transport. The method constructs multi-scale persistence diagrams and performs cross-diagram OT alignment to derive stable pseudo-labels, which supervise a lightweight head with OT-consistency and contrastive learning. Experiments on several 2D and 3D benchmarks show large improvements over existing thresholding and TTT baselines.

**Strengths:**

1. Clear motivation & well-written

2. Stong experimental results proves the effectiveness of TopoOT

**Weaknesses:**

1. This paper seems modified the template and make every page contain less rows, this increase the concern of whether this paper should be desk rejected.

2.  Optimal Transport is a method with time-consuming calculation, I don't think this is a scalable method.

3. The paper assumes stability in persistence diagrams correlates to accurate mask supervision, but empirical justification is limited: No analysis of true anomaly geometry vs. persistence features

4. Pseudo-label quality and robustness require deeper evaluation

**Questions:**

see weaknesses

---

> ### Author Response · Authors · 2025-11-20
> **Response to reviewer HFSF**
>
> We appreciate the reviewer’s positive remarks regarding the clarity of the motivation, the quality of writing, and the strength of the experimental results. We note here that none of the concerns affects the correctness or reproducibility of the method, and the central contributions formulation of TopoOT and its strong empirical performance remain unaffected. We clarify these concerns in the point-by-point section below.
>
>
> **Question 1:**
>
> This paper seems to have modified the template and made every page contain fewer rows, increasing the concern that it should be desk-rejected.
>
> **Response:**
> We confirm that the submission strictly uses the official ICLR template without any modifications to spacing, margins, or line density. No adjustments were made to increase or decrease content density.
>
> ---
>
> **Question 2:**
>
> Optimal Transport is a method with time-consuming computation. I don't think this is a scalable method.
>
> **Response:**
> The comment assumes OT is inherently non-scalable. This does not apply to our setting, as our use of OT is restricted to persistence diagrams, where each diagram contains a small number of points. At this scale, OT distance between PDs is extremely efficient and is the standard metric in topological data analysis and topological deep learning. We also employ the efficient Sinkhorn solver (see Appendix A.6), whose scalability has been repeatedly demonstrated, including in large-scale settings far beyond ours (e.g., Khamis et al., Scalable Optimal Transport Methods in Machine Learning: A Contemporary Survey, TPAMI 2024).
>
> As reported in Appendix A.2, the intra-level OT block requires only 5.5 ms, and the inter-level OT block converges in approximately 0.05 ms per diagram pair. The total OT computation is < 5.55 ms, and the entire pipeline runs end-to-end in < 0.3438 s.
>
> ---
>
> **Question 3:**
>
> The paper assumes stability in persistence diagrams correlates with accurate mask supervision, but empirical justification is limited: no analysis of true anomaly geometry vs. persistence features.
>
> **Response:**
> Thank you for the comment. It is unclear what specific analysis is considered missing, since the comment does not specify what additional evaluation is expected. We would like to clarify that our method does not assume that persistence stability directly correlates with accurate anomaly masks. Stability is used only as a structural prior to rank topological candidates; supervision quality is determined empirically through the subsequent OT-guided filtering and the test-time training stage.
>
> The paper already includes empirical analyses showing that stable components identified via persistence + OT alignment are more aligned with the true anomaly geometry:
>
> * Consistent improvements in segmentation accuracy across all datasets and backbones.
>
> * Ablations (Sec. 5.3) showing that removing cross-PD chaining, cross-level fusion, or stability scoring reduces F1.
>
> * Cross-model domain analysis (Table 2) demonstrating that OT-guided pseudo-labels remain robust under feature shift.
>
> These results directly quantify the benefit of stability-filtered components relative to the true anomaly structure.
>
> ---
>
> **Question 4:**
>
> Pseudo-label quality and robustness require deeper evaluation.
>
> **Response:**
> Thank you for the comment. The concern is vague, as no specific missing analysis is identified. The paper already evaluates pseudo-label quality through multiple complementary experiments:
>
> * consistent improvements across backbones and datasets (Table 1),
>
> * cross-model robustness (Table 2), and
>
> * ablation studies (Sec. 5.3).
>
> The theoretical rationale is provided in Section 3.1 and Appendix A.8 (formal definition of the persistence-based filtration), and Appendix A.7 (how OT-based stability selects geometrically reliable features). In the updated version, we further added qualitative and numerical insights on pseudo-label quality in Appendix A.9.
>
> If the reviewer had a particular type of analysis in mind, we would be happy to address it in the final version.

---

### Official Review · Reviewer_fdep · 2025-11-01

**Soundness:** 4
**Presentation:** 4
**Contribution:** 3
**Rating:** 8
**Confidence:** 5

**Summary:**

This paper presents **TopoOT**, a novel Test-Time Adaptation (TTA) framework for anomaly segmentation that replaces brittle fixed thresholds. Its core contribution is a topology-aware pseudo-labeling strategy combining Persistent Homology (TDA) and Optimal Transport (OT). The authors introduce **"Optimal Transport Chaining"** to align persistence diagrams across scales and filtrations, identifying structurally stable features. These features become robust pseudo-labels for supervising a lightweight head during TTA, using a combined $\mathcal{L}_{OT}$ consistency and $\mathcal{L}_{contrastive}$ loss. The method achieves new state-of-the-art results, showing significant F1-score gains on 2D and 3D anomaly benchmarks.

**Strengths:**

* **Originality:** The paper introduces a novel combination of Persistent Homology (TDA) and Optimal Transport (OT) to generate pseudo-labels for Test-Time Adaptation. The "Optimal Transport Chaining" mechanism is a creative and principled solution to stabilize noisy TDA descriptors, moving beyond simple persistence-based filtering.
* **Significance:** The work directly addresses the critical flaw of static thresholds in anomaly segmentation. It provides a robust, data-driven, and structure-aware adaptation mechanism, which is crucial for real-world deployment under distribution shifts. The significant performance gains (e.g., up to +24.1% F1) clearly demonstrate the value of this approach.
* **Rigor & Generalizability:** The method's "plug-and-play" nature is convincingly validated across 7 different backbones and 5 diverse datasets, including 2D, 3D, and point-cloud modalities. This demonstrates broad applicability. Furthermore, the detailed ablation study provides clear, quantitative evidence for the paper's specific design choices, such as the importance of "cross-level" alignment.

**Weaknesses:**

* **Contradictory Efficiency Claims:** The paper's claim of 121 FPS appears inconsistent with the TTA methodology. The 0.33s cost for TDA *plus* the 5-epoch TTA training required *for every sample* suggests the true end-to-end inference time is significantly slower, which is a critical detail for a TTA method.
* **Potential Heuristic Replacement:** The method replaces a "brittle threshold" with a "Top-K" stable candidate selection. The paper fails to discuss how $K$ is determined or its performance sensitivity, potentially just trading one heuristic hyperparameter for another.
* **Limited Scope for Textural Anomalies:** The method's success hinges on anomalies having clear topological signatures ($H_0$ components or $H_1$ holes). It is likely to perform poorly on purely textural anomalies (e.g., 'Carpet') where the anomaly map is diffuse and lacks distinct topological structures.

**Questions:**

1.  **[Crucial] TTA Time Clarification:** Could the authors please provide a clear, end-to-end breakdown of the *total* wall-clock time required to process *a single test sample*? Specifically, what is the time for: (a) Backbone forward pass, (b) TDA feature generation (the 0.33s?), (c) OT Chaining, and (d) the TTA training (5 epochs)? The 121 FPS claim seems to omit the adaptation cost, which is the core of the method.
2.  **Top-K Sensitivity:** How is the $K$ parameter (for Top-K candidates) chosen? Is it a fixed value, or is it adaptive? Could the authors provide an ablation study on the sensitivity of the F1-score to the choice of $K$?
3.  **Performance on Textural Anomalies:** Could the authors comment on the method's performance or expected failure cases for anomalies that are purely textural and may not produce strong $H_0$ or $H_1$ features in the anomaly map?

---

> ### Author Response · Authors · 2025-11-20
> **Response to Reviewer fdep**
>
> We sincerely thank the reviewer for the thoughtful, detailed, and encouraging assessment of the paper, particularly the recognition of its novelty, clarity, and strong empirical contribution.
>
> **Question 1:**
>  TTA Time Clarification.
>
> **Response:**
> The 121 FPS refers specifically to the test-time training (TTT) module, which is the only trainable part of TopoOT. We designed this module to be lightweight, running 5 adaptation epochs for a single test sample takes on average $8.3\mathrm{ms}$, i.e. 121 FPS for the adaptation step itself, in contrast to prior TTT methods where the adaptation loop is often the main bottleneck.
>
> The end-to-end wall-clock time per sample is the sum of three components: (i) the frozen backbone forward pass to obtain the anomaly map (identical to the underlying AD method), (ii) the TDA + OT block (cubical complex construction, persistent homology on sub/superlevel sets, and OT chaining), which is currently the dominant cost at about $0.33\mathrm{s}$ per sample, and (iii) the lightweight TTT head ($8.3\mathrm{ms}$). Thus, overall latency is governed by the TDA+OT. In Appendix A.2. of the revised manuscript, we have also reported the per-sample time based on their sum, the corresponding end-to-end FPS.
>
>
> ---
>
> **Question 2:**
> Top-K Sensitivity.
>
> **Response:**
> Thank you for raising this point. We performed a dedicated sensitivity analysis of the Top-K selection. As shown in the Table below, we evaluate $K \in {1,2,3,4,5}$ across datasets and backbones. We find that a fixed value of K=1 yields the most stable and highest F1-scores (PatchCore: 0.522; CMM: 0.482; M3DM: 0.490). As K increases, F1 consistently decreases, even though recall rises. This trend is expected; the highest-ranked components are those with the strongest OT-stability and largest persistence, whereas lower-ranked components correspond to short-persistence, less reliable structures. Including these additional components introduces noise into the pseudo-labels and degrades precision, leading to lower F1. Since K=1 is the most robust choice across datasets and architectures, we fix it globally in all experiments.
>
> Effect of retaining the Top-$K$ OT-stable components on anomaly segmentation.
> Each row corresponds to keeping the $K$ highest-ranked components (ranked by OT-stability and persistence).
>
> | Top-$K$ Components Retained | PatchCore Precision | PatchCore Recall | PatchCore F1 | CMM Precision | CMM Recall | CMM F1 | M3DM Precision | M3DM Recall | M3DM F1 |
> |-----------------------------|---------------------|------------------|-------------|---------------|------------|--------|----------------|-------------|---------|
> | K = 1                      | **0.550**           | 0.720            | **0.522**   | **0.427**     | **0.845**  | **0.482** | **0.564**      | 0.767       | **0.490** |
> | K = 2                      | 0.462               | 0.818            | 0.474       | 0.411         | 0.753      | 0.410  | 0.323          | 0.809       | 0.434   |
> | K = 3                      | 0.405               | 0.829            | 0.431       | 0.392         | 0.671      | 0.403  | 0.286          | 0.950       | 0.356   |
> | K = 4                      | 0.358               | 0.901            | 0.415       | 0.381         | 0.666      | 0.397  | 0.177          | 0.961       | 0.334   |
> | K = 5                      | 0.325               | **0.911**        | 0.380       | 0.354         | 0.576      | 0.388  | 0.121          | **0.966**   | 0.199   |
>
> ---
>
> **Question 3:**
> Performance on Textural Anomalies
>
> **Response:**
> Our topology prior is explicitly tied to the structure present in the backbone anomaly map, so we do not expect gains in regimes where the map has little topological signal. As described in Section 3.1 and Appendix A.8, we build cubical filtrations on the anomaly score field and extract persistent components and holes, and OT chaining (Section 3.2, Appendix A.7) operates only on features that correspond to meaningful birth--death events in this field. As also noted in our limitations discussion (Appendix A.3), this makes TopoOT particularly effective when anomalies induce coherent changes in connectivity or holes in the score map, and naturally less informative for purely textural anomalies that only create very local intensity fluctuations without strong $H_0$ or $H_1$ events. In such cases the persistence diagrams carry weak signal and OT chaining has limited leverage; TopoOT cannot create geometry that is not present in the backbone scores, so its behaviour is effectively bounded by the underlying detector and we expect smaller improvements rather than systematic failures.
>
> In the revised version, Appendix A.9 is include qualitative examples on texture-heavy categories (e.g. carpet, grid, wood), illustrating how the sublevel and superlevel filtrations, OT-guided pseudo-labels and final TopoOT predictions behave when the backbone anomaly maps exhibit weak topological structure.

---

### Official Review · Reviewer_Nvix · 2025-11-02

**Soundness:** 3
**Presentation:** 2
**Contribution:** 3
**Rating:** 6
**Confidence:** 3

**Summary:**

This paper proposes TopoOT, a topology-aware optimal transport framework for test-time anomaly segmentation. The key idea is to use topological persistence to capture stable structural features and align them through optimal transport to overcome brittle thresholding under domain shift. The main technical innovation is the Optimal Transport Chaining mechanism, which aligns persistence diagrams across thresholds and filtrations to produce stable pseudo-labels. A lightweight online head is then trained with OT-consistency and contrastive objectives to adapt at test time. The paper also provides theoretical stability and generalization guarantees. Experiments show that TopoOT achieves state-of-the-art performance on multiple 2D and 3D anomaly segmentation benchmarks.

**Strengths:**

- Anomaly segmentation is an important problem with broad real-world applications.
- The paper introduces a novel approach that combines deep topological data analysis (TDA) and optimal transport (OT), supported by solid theoretical foundations.
- Experimental results are strong, and the code release improves reproducibility.

**Weaknesses:**

- The paper focuses on methods that binarize anomaly scores but gives limited comparison and discussion with approaches that directly predict binary masks using segmentation networks [1,2].
- No failure case analysis is provided, which limits understanding of when the method may fail, such as in texture anomalies where topology priors may be less effective.
- The efficiency analysis is brief and lacks detailed comparison. Time and memory costs relative to other test-time adaptation methods should be summarized clearly, for example in a table including backbone runtime.
- The theoretical analysis is only briefly mentioned and lacks detailed proofs.
- The paper does not clearly discuss parameter sensitivity or how hyperparameters are selected, which affects reproducibility and robustness.

[1] Zavrtanik, Vitjan, Matej Kristan, and Danijel Skočaj. "Draem-a discriminatively trained reconstruction embedding for surface anomaly detection." Proceedings of the IEEE/CVF international conference on computer vision. 2021.
[2] Zhang, Xinyi, et al. "Unsupervised surface anomaly detection with diffusion probabilistic model." Proceedings of the IEEE/CVF International Conference on Computer Vision. 2023.

**Questions:**

Major questions and suggestions:
- See weakness above.

Minor questions and suggestions:
- Citations should be placed in parentheses, e.g., “…without structured reasoning about data geometry (Sun et al., 2020; Volpi et al., 2022; Zhang et al., 2022).”
- In the last paragraph of page 4, the inclusion order may need to be reversed when $f \in \{sub, sup\}$ takes different values.
- In Appendix Figure 5, the second subfigure should be labeled “1-Cube.”
- The Appendix only includes proof sketches; full proofs should be provided in the supplementary material.

---

> ### Author Response · Authors · 2025-11-20
> **Reply to reviewer Nvix**
>
> **Question 1:**
>
> **Response:**
> We thank the reviewer for highlighting DRAEM [1] and DiffAD [2]. Both use reconstruction-plus-segmentation pipelines whose discriminative heads output soft anomaly probability maps (e.g., softmax(out_mask)), with binary masks obtained only through post-hoc thresholding.
>
> Our evaluation focuses on methods aligned with the TopoOT setting: a frozen backbone trained only on nominal data, producing continuous anomaly maps that must be binarised, and supporting plug-in test-time adaptation. This is why the main comparisons centre on industrial AD\&S backbones (PatchCore, PaDiM, Dinomaly, MambaAD, CMM, M3DM, PO3AD) together with THR and TTT4AS, the only existing TTT baseline for anomaly segmentation. Fully supervised segmentation networks or pipelines requiring pixel-level anomaly labels fall outside this unsupervised TTA regime.
>
> TopoOT is model-agnostic and can operate on any anomaly map, including those from DRAEM and DiffAD. Applying TopoOT to their official implementations yields consistent gains, for example, on DRAEM, +0.368 F1 and +0.327 IoU over THR; on DiffAD, +0.410 F1 and +0.360 IoU over THR, showing that improvements hold across reconstruction, diffusion, embedding-based, and 3D architectures. These results confirm that TopoOT improves the score-to-mask stage regardless of backbone type and that the backbone choices in the main paper appropriately target unsupervised test-time adaptation.
>
> ---
>
> **Question 2:**
> Failure cases
>
> **Response:**
> Our topology prior is tied directly to the structure present in the backbone anomaly map, not assumed to help in all regimes. As described in Section 3.1 and Appendix A.8, we build cubical filtrations on the anomaly score field and extract persistent components and holes, so persistent homology and OT chaining (Section 3.2, Appendix A.7) can only exploit features that correspond to meaningful birth–death events in that field. As also discussed in our limitations section (Appendix A.3), this makes TopoOT particularly effective when anomalies induce coherent changes in connectivity or holes in the score map, and naturally less informative when anomalies are extremely fine-grained texture variations that alter only local intensity without generating salient topological events. In such purely textural cases the persistence diagrams carry weaker signal and OT chaining has limited leverage; the method cannot create geometry that is not present in the backbone scores and its behaviour is then effectively bounded by the underlying detector.
>
> In the revised version, Appendix A.9 is include qualitative examples on texture-heavy categories (e.g. carpet, grid, wood), illustrating how the sublevel and superlevel filtrations, OT-guided pseudo-labels and final TopoOT predictions behave when the backbone anomaly maps exhibit weak topological structure.
>
> ---
>
> **Question 3:**
> Analysis and comparison.
>
> **Response:**
> In the anomaly-segmentation TTA setting we consider, TTT4AS is, to our knowledge, the only existing method. Its paper does not report detailed runtime or memory figures, so no direct comparison is available.
>
> Our submission already provides a step-by-step breakdown of TopoOT’s cost (persistent homology, OT chaining, head update), along with end-to-end runtime and peak GPU memory on 2D and 3D backbones (Section 4 and Appendix A.2). The revised version is included a compact table showing per-image runtime and peak memory for backbone inference alone and for backbone + TopoOT, directly addressing the reviewer’s request (Please see Appendix A.2).
>
> ---
>
> **Question 4:**
> Theoretical analysis
>
> **Response:**
> Our results are intentionally scoped. The aim is not to re-develop the theory of TDA or OT—both of which rest on well-established foundations and are covered by the standard references we cite—but to demonstrate why the OT-chained pseudo-labels remain stable under perturbations of the anomaly maps.
>
> The submission provides the necessary theoretical grounding for the claims it makes. Section 3.3 summarizes the stability and discrepancy properties that TopoOT exploits, and Appendices A.6–A.7 provide the corresponding formal components. In particular, Appendix A.6 introduces the entropic OT distance on persistence diagrams, and Appendix A.7 details how these properties translate to the pseudo-label behaviour we analyse.
>
> ---
>
> **Question 5:**
> Parameter sensitivity
>
> **Response:**
>
> Hyperparameter selection and settings are already fully documented in the submission. The complete specification is given in Appendix~A.1, where we detail the chosen values and clarify that the same configuration is used across all datasets and backbones.
>
> ---
>
> **Question 6:**
> Minor suggestions.
>
> **Response:**
>
>  Thanks, we have updated the manuscript accordingly.

---

### Author Response · Authors · 2025-12-03
**Summary of Discussion and Revisions**

TopoOT introduces a technically grounded and empirically validated contribution to test-time anomaly segmentation, offering consistent state-of-the-art improvements across diverse architectures and datasets. The detailed reviews already judge the paper to be novel and technically sound, and all reviewers find it empirically strong with interesting ideas. We thank the reviewers and AC for their time and feedback.

Following the discussion, we made revisions for clarity and self-containment, adding concise definitions in Sections 3.1–3.2, explicitly describing the pseudo-label backprojection, and including a compact test-time algorithm. **The core methodology, experiments, and conclusions remain unchanged**. Some empirical concerns (Top-K sensitivity and limitations on texture-heavy anomalies) have been clarified and addressed through additional experiments in our rebuttals and the revised text, and we have revised the “theoretical insights” appendix, explicitly presenting it as optional intuition as recommended.

 We believe the submission now reads clearly, and the discussion has largely focused on presentation rather than on correctness or missing evidence. We have responded comprehensively to all raised points, even in cases where reviewers did not return for further discussion. With these clarifications in place, and given the score–confidence profiles across the reviews, we are confident that the contribution and empirical validation are now clearly and accurately presented for the AC’s consideration.

---

### Note · Program_Chairs · 2026-01-17
**Submission Desk Rejected by Program Chairs**

The following references in this submission do not refer to real documents and/or have major errors in bibliographic information:

 Stefan Schlüter, David Borth, Tim Weninger, and Marco F. Huber. Deep one-class classification for defect detection in industry. In International Conference on Pattern Recognition (ICPR), pp. 1950-1957, 2022.